

# Hydrological functioning of West-African inland valleys explored with a critical zone model

Basile Hector[1], Jean-Martial Cohard[1], Luc Séguis[2], Sylvie Galle[1], Christophe Peugeot[2]

[1]*IGE*, Univ. Grenoble Alpes, CNRS, IRD, Grenoble INP, F-38000 Grenoble, France

[2]*HSM*, Univ. Montpellier, CNRS, IRD, F-34000 Montpellier, France
*Correspondence to*: Basile Hector (basile.hector@ird.fr)

**Abstract.** Inland valleys are seasonally waterlogged headwater wetlands, widespread across West-Africa. Their role in the
hydrological cycle in the humid, hard rock-dominated Sudanian area is not yet well understood. Thus, while in the region
recurrent floods are a major issue, and hydropower has been recognized as an important development pathway, the scientific
community lacks a precise knowledge of streamflow (Q) generation processes and how they could be affected by the presence
of inland valleys. Furthermore, inland valleys carry an important agronomic potential, and with the strong demographic rates
of the region, they are highly subjected to undergo land cover changes. We address both the questions of the hydrological
functioning of inland valleys in the Sudanian area of West-Africa and the impact of land cover changes on these systems
through deterministic sensitivity experiments using a physically-based critical zone model (ParFlow-CLM) applied on a
synthetic catchment which comprises an inland valley. Model forcings are based on 20 years data from the AMMA-CATCH
observation service and parameters are evaluated against multiple field data (Q, evapotranspiration –ET-, soil moisture, water
table levels, and water storage). The hydrological model applied to the conceptual lithological/pedological model proposed in
this study reproduces the main behaviors observed on a highly instrumented elementary catchment. We found that yearly water
budgets were highly sensitive to the vegetation distribution: average yearly ET for a tree-covered catchment (944 mm) exceeds
that of herbaceous-cover (791 mm). ET differences between the two covers vary between 12 and 24 % of the precipitation of
the year for the wettest and driest year, respectively. As a consequence, the tree-covered catchment produces a yearly Q budget
28 % lower on average as compared to a herbaceous-covered catchment, ranging from 20 % for the wettest year to 47 % for a
25 dry year. Trees also buffer interannual variability in ET by 26 %. On the other hand, pedological features (presence -or absence-
of the low permeability layer commonly found below inland valley, upstream and lateral contributive areas) had limited impact
on yearly water budgets but marked consequences on intraseasonal hydrological processes (sustained/unsustained baseflow in
the dry season, catchment water storage redistribution…). Therefore, subsurface features of inland valleys have potentially
significant impacts on downstream water-dependent ecosystems and water uses as hydropower generation, and should focus
30 our attention.



## 1 Introduction

It is widely recognized that wetlands exert a strong influence on the hydrological cycle. However, they hardly support generalization in their functioning, according to their different types and settings, and the limited available hydrological observations (Bullock and Acreman, 2003). In Sub-Saharan West-Africa, "Bas-fonds", or inland valleys, are seasonally

waterlogged wetlands located at the headwater of streams and cover 22-52 Mha (Andriesse et al., 1994). They can be found in the whole continent, although they hold different names, such as dambos, mbuga, inland valleys, or valley bottoms. Their role in the hydrological cycle is still unclear and lacks significant attention. This is particularly true for the humid Sudanian zone, mostly located on hard-rock lithology with rather shallow water tables and low-capacity aquifers (MacDonald et al., 2012) where all components of the critical zone are tightly linked (e.g. trees roots systems are likely to be connected to the water

tables (Mamadou et al., 2016; Richard et al., 2013)). There, for instance, the role of inland valleys in streamflow generation is still poorly understood, while they are suspected to provide most of the observed streamflow as baseflow through exfiltration of perched seasonal water tables (Séguis et al., 2011).

Streamflow generation mechanisms deserve more attention, as (1) the region suffers severe floods, which affect an increasing number of people (Di Baldassarre et al., 2010) and (2) hydropower development is also undergoing (Cervigni et al., 2015) and

is recognized to be a fundamental large scale source of energy for the development of Sub-Saharan Africa (http://www.worldbank.org/en/topic/hydropower/overview, April 2018) (World Bank, 2009).

Moreover, the strong West-African demographic rates (2.6 % in the period 2000-2010) imply rapid changes in land use and land cover (LULC) (FAO, 2012). For instance, about 42 % (respectively 8 %) of forested land (respectively non-forested vegetation) have been converted into agricultural land (which amount to an increase of more than 11 %) in the Sudanian area

between 1975 and 2000 (Eva et al., 2006). This is also likely to affect the functioning of inland valleys, although it is yet unknown to what extent.

Despite the possible role of inland valleys in the hydrological cycle, they carry an important agronomic potential currently underexploited (Katic et al., 2014; Rodenburg et al., 2014). Due to their high water content, they can be cultivated offseason, and thus buffer the high climatic variability of the area (Nyamadzawo et al., 2015), as well as human pressure on uphill plots

located closer to the villages (Lounang Tchatchouang et al., 2014). However, inland valley exploitation limits should be carefully studied, as over-exploitation may have consequences on these specific ecosystems (Böhme et al., 2016; Brouwer et al., 2014; Wood, 2006). Rodenburg et al. (2014) estimate that production derived from less than 10 % of the total inland valley area over Africa could meet the total current demand for rice in Africa. In these studies, little is mentioned about water exploitation sustainability. This is due to the heterogeneity of inland valleys and the difficulty to capture fundamental

hydrological processes to be generalized. This again calls for documenting water redistribution processes within inland valleys, so as to include water use sustainability in the process of inland valleys selection and cultivation (Böhme et al., 2016; Schmitter et al., 2015).





Headwater inland valleys are characterized by a rapid decrease of permeability with depth due to the accumulation of clays, which is ultimately associated to a low permeability layer, allowing the formation of perched water tables in the valley thalweg (Blavet, 1997; Brabant, 1991; Hector et al., 2015; von der Heyden and New, 2003; Von Der Heyden, 2004). The hydrology of inland valleys is also controlled by classical catchment features like vegetation distribution and the functioning of upstream

contributive area (Von Der Heyden, 2004). For instance, Balek (2006) reports on increased flood flow when trees were removed from two of their observed catchments. The fast LULC changes occurring today call the scientific community to clearly understand the role of different vegetation distribution and the impact of their changes on the hydrology of inland valleys. These interactions between flow generation and vegetation require studying the hydrological functioning of inland valleys in an integrated, critical zone (CZ)-like approach.

In complement of measurements of hydrological fluxes (e.g. precipitation, streamflow, evapotranspiration) and storage variables (soil moisture, water table levels, storage), hydrodynamic properties and geophysical imagery and monitoring, physically-based CZ models are appropriate tools to study the functioning of a hydrosystem and the relationships between the components of its CZ (e.g. Kollet and Maxwell (2008); Srivastava et al. (2014)), by conducting virtual experiments (Weiler and McDonnell, 2004). These models explicitly link all hydrological compartments of the CZ, from the deep impermeable

bedrock to the top of canopy.

In this paper, we seek to address the two following questions: 1) what are the main characteristics of the hydrological functioning of inland valleys in the Sudanian area of West Africa? 2) What is the impact of LULC changes on such systems? The approach undertaken here is to build a set of virtual experiments in an idealized elementary V-shaped catchment to test the sensitivity of a physically-based CZ model to different features: a) the presence of the low permeability layer in the valley

thalweg, which characterizes inland valleys, b) the pedology of contributive areas and c) the vegetation distribution (from natural woodland cover to herbaceous-like fallow/crops cover). This deterministic modeling approach largely builds on the large panel of observations available within the AMMA-CATCH observation service (www.amma-catch.org (Galle et al., Submitted; Lebel et al., 2009) and several campaigns, as well as on a highly instrumented elementary catchment for which we previously built a conceptual lithological/pedological model (Hector et al., 2015).

In the first section, we briefly discuss the physical environment and how we model it using a physically-based CZ model. Then, we present the results of a reference case, which are compared to observations from an elementary headwater catchment in the hard-rock area of the Sudanian region, to show that the model is able to reproduce to a large extent the complex critical zone behavior. We finally use the results of a set a virtual experiments to infer the model sensitivity to the main inland valley features (presence of a clay layer, hydrodynamic properties of the contributive areas, vegetation distribution), and discuss these

results.

## 2 Material and methods

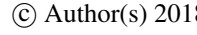



## 2.1 Description of the physical environment

In this section, we briefly describe our conceptual representation of the main soil and vegetation characteristics found in the hard-rock basement areas of the Sudanian climatic region (yearly precipitation amount between 700 and 1400 mm). This eco-physical model is built following the literature and the work of (Hector et al., 2015), who studied and described the functioning

of an elementary catchment in North Benin, the Nalohou catchment (0.16 km²), which is part of the regional AMMA-CATCH observatory. See for instance their Fig. 13 for a conceptual sketch of their understanding of the Nalohou catchment: during the dry season the permanent water table is disconnected from the inland valley, while during the wet season, the permanent water table rises and a shallow perched water table appears in the inland valley which is fed by both rainwater infiltration and lateral subsurface transfers. Lateral subsurface contributions to the inland valley differ from each bank: contributions from the layered

left bank (highly permeable hard-pan over low permeability clay-rich horizons) exceed contributions from the sandier right bank. Fig. 1.a shows the actual catchment shape as a black line and cross sections of Fig. 1.b-c summarize the internal structure of the catchment.

### 2.1.1 Soil layers and properties

Our conceptual model of the physical environment is made of three layers through depths (H1: 0-0.5 m, H2: 0.5-2 m, H3: 2 m -10.5 m, Fig. 1.b-c), to which we prescribed specific hydrodynamic properties in a deterministic way and following the literature, but also based on numerous field and laboratory measurements by Hector (2014); Richard (2014) and Robert (2012) in this Sudanian area (Table 1). In the following text, we refer to sand-like and clay-like hydrodynamic properties in a broad sense, of respectively highly permeable/low retention media and low permeable/high retention media, but the reader may refer

to Table 1 and Table 3 for actual properties ranges and values.

The thicknesses of the two shallowest soil layers (H1 & H2) are well defined from observations in the Nalohou catchment, and seem in accordance with other locations in (sub)tropical humid regions (de Condappa et al., 2008). The deeper layer (H3) thickness is the regolith thickness, located above the fissured basement and the hard rock basement, following (Lachassagne et al., 2011).

25                                        Figure 1

*Layer H1:*

In the tropical-humid Sudanian climate, top soils are usually permeable, due to microfauna bioturbation, or soil work from farmers (Giertz and Diekkrüger, 2003; Richard, 2012; Robert, 2012). As a consequence, the infiltration-excess overland flow is very limited in the area (e.g., Giertz et al. (2006); Masiyandima et al. (2003). The top-soil layer, H1, is therefore permeable,

with sand-like soil hydrodynamic properties. Some authors note a significant relationship between saturated hydraulic conductivity and land cover in the area (Giertz et al., 2005; Giertz and Diekkrüger, 2003; Richard, 2014), yet in a limited range (less than an order of magnitude). The H1 layer is thus considered laterally homogeneous in this study.



*Layer H2:*

The second layer (H2) is usually more variable laterally, and variations in clay content in this potentially illuviation layer (a clay-accumulation horizon, as described by de Condappa et al. (2008)) may generate perched water tables (e.g. Giertz et al., (2006)). In this layer, some macroporous hardpan units with high permeability may also be found (Bonsor et al., 2014; Hector et al., 2015).

It is beyond the scope of this study to analyse the effect of this layer (H2) on the hydrological processes, because it would likely necessitate an entire study. We therefore chose to consider a rather sandy, permeable layer, to mimic a hardpan unit behavior (as part of the polycyclic soil profile described by Faure and Volkoff (1998)), and favor quick lateral subsurface transfer, which have been evidenced in the area (e.g Hector et al. (2015); Masiyandima et al. (2003); Séguis et al. (2011)).

*Layer H3:*

The composition of the third layer (H3) is variable laterally. In this study, we are interested in two extreme cases of what can be found in this regolith layer, according to Faure and Volkoff (1998). On the one hand, there is a low-permeability, clay-accumulation-like horizon, similar to what is observed by Hector et al. (2015); Wubda et al. (2017) in the Western half of the Nalohou catchment, and also described as a major feature from larger scales observations by Faure and Volkoff (1998). On the other hand, there is a clay-eluviated, saprolite-like, sandy horizon, which is expected to produce a strongly contrasted behavior. It is also similar to what is observed on the Eastern half of the Nalohou catchment, and being the second major feature from larger scales observations by Faure and Volkoff (1998). Anticipating on the following sections, the two different cases for H3 layer (clay-like and sand-like saprolite) will be evaluated in the virtual experiments, together with an alternative case where the H3 layer in the left bank of the virtual V-shape catchment will be clay-like and the right bank will be sand-like. In the 'reference' simulation, H3 will have clayey hydrodynamic properties, in the 'saprolite' simulation it will have sandy hydrodynamic properties and in the 'mixed' simulation it will have clayey (resp. sandy) hydrodynamic properties in the West (resp. East) bank.

*Inland valley:*

The inland valley is located in the downstream part of the valley thalweg (circles of Fig. 1.a). It has a specific H1/H2/H3 sequence. It usually consists of sandy upper horizons. H1 and H2 layers are the same as elsewhere, and are considered to be of sandy nature and to favor quick vertical and lateral transfers, as observed in the field. At about 2m depth, in the upper-part of the H3 layer just below the H2 horizon is a low permeability, clayey layer, which actually characterizes the inland valley. This allows the formation of perched water tables during the rainy season (Masiyandima et al., 2003). Such shallow drainage system is the main contributor to baseflow, which represent roughly about 80% of total streamflow for such catchment (Giertz et al., 2006; Hector et al., 2015; Kamagaté et al., 2007; Séguis et al., 2011). This clay-rich accumulation horizon is often found above a poorly weathered horizon (Brabant, 1991), as is the case in the Nalohou catchment. Therefore the H3 layer underneath the inland valley is made of low-retention, high-permeability fissured basement (in the sense of Lachassagne et al. (2011)).

*Hydrodynamic properties:*





In a thesis work, Hector (2014) summarized the variability of field measurements to build up classes of main soil/lithology units for the three main layers identified in this study (Table 1). Hydrodynamic parameters (following Van Genuchten (1980)) for this study are shown below on Table 3. Saturated conductivities of saprolite and fissured basement in the H3 is given by the range of observations derived both from pumping tests and magnetic resonance soundings in the area (Vouillamoz et al., 5 2014).

Table 1

### 2.1.2 Vegetation

Non-disturbed natural vegetation in this area is mostly woodland, composed of trees with an herbaceous layer underneath. Under human influences, trees are usually cut in favor of agricultural plots. However trees in the riparian areas and the inland 10 valley fringes are generally kept alive, as well as specific species on the hillslopes for consumption purposes (agroforestry system). This is why we consider in this study two vegetation classes: trees and herbaceous. While the trees class refers to a fully tree-covered state with some herbaceous layer underneath, the herbaceous cover refers to a mixed fallow-field landscape with sparse trees, composed of either cultivated plots or young fallows under the influence of farmers, and with some isolated trees (with height <10 m) kept for local consumption.

15 Trees phenology mostly consists in defoliation around January, which occurs asynchronously between different species and different individuals of the same species (Seghieri et al., 2012). On the other hand, in the so-called herbaceous cover, the ground is commonly covered by scattered shrubs, while the actual herbaceous layer usually starts growing in April after the first rains and reaches its maximum height (up to 2.5 m) in October. At this time its height exceeds that of shrubs. Then, the herbaceous layer stops transpiring and dries up from October to January. Controlled fire is a traditional practice in Benin; the 20 vegetation (mainly the herbaceous layer) is burnt between November and February, when the soil and vegetation are dry. Hence, during the dry season, the soil is bare for about five months. Leaf area index (LAI) and Stem area index of trees and herbaceous classes are later shown on Fig. 6.a.

### 2.2 Critical zone model

The CZ model ParFlow-CLM (PFCLM) is composed by ParFlow, a three-dimensional integrated physical model that 25 simultaneously solves variably saturated flow and surface outflow coupled with CLM, a land surface model which solves both the water and energy budget at the surface. ParFlow uses a free-surface overland flow boundary condition which routes ponded water as overland flow using the kinematic wave equation and Manning equation (Kollet and Maxwell, 2006). In this study, the topographic effect is taken into account by a terrain following grid (Maxwell, 2013), and only requires two-dimensional surface slopes to be provided. ParFlow solves the Richards equation using cell-centered finite differences in space and an 30 implicit backwards Euler scheme in time. The Newton-Krylov method is used together with a multigrid preconditioner and shows excellent parallel scaling, which allows the simulation of large scale domains (i.e. continental US) at high resolution (i.e. < 1km) (Ashby and Falgout, 1996; Jones and Woodward, 2001; Maxwell et al., 2014a). ParFlow is coupled through the



user-defined *N*-first cells from the surface, to a modified version of the Community Land Model (CLM) which solves both the water and energy budget at the surface (Dai et al., 2003; Maxwell and Miller, 2005). CLM has one vegetation layer and the version within ParFlow currently only allows a single vegetation class per surface grid cell. PFCLM is an excellent candidate for this study because it allows to represent a number of physical processes that have been observed in Sudanian West Africa

(perched water tables, groundwater and subsurface water baseflow, stormflow, plant transpiration and bare ground evaporation…). This is achieved using physical parameters (hydrodynamic properties, vegetation parameters, complex 3-dimensional lithological or pedological units…) and without prior specification of the processes. The model has already been successfully applied in the region, using measured parameters for simulating vertical water fluxes (evapotranspiration, percolation) and specific features like perched water tables (Robert, 2012).

**2.3 Model setup**

We model an idealized elementary headwater catchment, comprising an inland valley, of 720m x 300m catchment (0.216 km²), with a 20m lateral (x,y) resolution (Fig. 1, Table 2). The idealized inland valley and its contributive area is a V-shape type (Maxwell et al., 2014b), sloping in the Y direction (Fig. 1.a). Y-direction slope is 4 % and X-direction slope is 2 %. The cells in the center represent the potentially overflowing section of the inland valley. These cells are further lowered by 2 m below

the altitude of the inland valley fringes, to represent the classically observed banks.

Table 2

The vertical resolution varies with depth following a geometric progression of common ratio 1.2 for the first 13 cells (which are coupled to CLM), ranging between 0.134 m to 1.195 m and four more 1 m-thick cells, uncoupled to CLM, are added at the bottom. The domain is therefore 10.5 m thick. Selected hydrodynamic parameters based on expertise-driven sensitivity studies

within the range of observed values (Table 1) are shown on Table 3.

Table 3

No-flow boundary conditions are applied on the catchment edges. The only way for the water to leave the domain, apart from evapotranspiration, is through overland flow at the catchment outlet.

**2.4 Model forcings**

**2.4.1 Atmospheric forcing**

Forcing data is obtained from the AMMA-CATCH observatory (Lebel et al., 2009). The meso-scale site of the upper Ouémé catchment (14300 km²) in northern Benin is representative of the Sudanian climate. The meteorological variables used to force CLM (wind speed, air temperature and humidity, pressure and the incoming radiative long and short waves, Fig. 2) were measured at a 2 m height on a flux tower located on a cultivated site (the Nalohou catchment, lat. 9.74∘N, long. 1.60∘E, 449 m

a.s.l). Rainfall is measured with a tipping-bucket rain gauge.



West African rainfall follows the monsoon dynamics and is characterized by a strong variability at interannual scales (see e.g. Lawin (2007); Le Lay and Galle (2005)) for detailed analyzes of the Sudanian rainfall regime). On the simulated years, yearly precipitation amounts range from 913 mm to 1572 mm (Fig. 2).

Fig. 2 shows the meteorological variables used to force PFCLM. They have been described in Mamadou et al. (2014) for the year 2008 and recently compared to data from a 13 km-away site over a forested cover for the period 2008-2010 (Mamadou et al., 2016). The daily meteorological conditions of the two sites were expectedly close. Minimal values of incoming short waves radiations (SWin) are observed during the wet season due to cloud cover, and low values are also found at the core of the dry season, due to aerosols brought by the north-easterly Harmattan winds. Incoming long wave radiations (LWin) are low during the dry season due to the reduced cloud cover. Highest air temperatures are observed before the start of the wet season, while the wet season exhibits the lowest daily average and amplitude of temperature. Specific humidity clearly shows the monsoon/Harmattan signal: during the wet season, moist air is brought by the south-westerly to southerly monsoon winds and during the dry season north-easterly Harmattan winds bring dry air from the Sahara. The Harmattan blows as a light to moderate breeze. These weather types are typical for Sudanian regions (Lohou et al., 2010; Sultan and Janicot, 2003).

PFCLM is being forced at a 30 mn time step, over the 7-years period 2006-2012. Initial conditions are set as a water table depth of 3 m, in accordance with several water table measurements. The model is spun up for one year which corresponds to the year 2007, a rather average year.

Figure 2

### 2.4.2 Vegetation forcings

In this study, we forced the model with observation-based time series of LAI (Leaf Area Index), SAI (Stem Area Index), displacement heights and roughness lengths for the two vegetation classes of our domain (trees and herbaceous).

Vegetation height was monitored over an herbaceous-covered site (Nalohou) and a forest site (Bellefoungou) 13 km away and further used to calibrate a simple API (Antecedent Precipitation Index)-like model which takes precipitation as input (Kohler and Linsley, 1951), and with the assumption that vegetation growth follows soil moisture availability. This allowed deriving vegetation heights during the growing period at daily time step for herbaceous cover for years where only precipitation data was available. Roughness lengths and displacement heights have been derived from these time series following Brutsaert approach (Brutsaert, 1982) calibrated on in-situ eddy covariance measurements. The LAI time series is a combination of satellite LAI products (CYCLOPE, MODIS, SEVIRI), constrained by in situ measurements derived from hemispherical photographs based on the method proposed by Weiss et al. (2004), both on the forested and herbaceous-covered site. Herbaceous SAI is also calculated from LAI in respect *with (LAI + SAI = $LAI_{max}$)* during the senescence period (See later Fig. 6.a for the LAI and SAI time series used in the model forcing).





For the herbaceous cover, the root distribution follows that of Zeng (2001), which is an exponential shape for the cumulative root fraction (Y) to depth (d) of equation $Y = 1 - \frac{1}{2}(e^{-ad} + e^{-bd})$, with the two coefficients a= 9 and b=2. For the trees, the distribution is kept uniform over the first three meters below the surface, because we found the Zeng equation not relevant for most Sudanian trees for which efficient deep roots have been observed (Jourdan, Gnanglé, Seghieri and Peugeot, pers.

Communication). The closest published study on deep roots in natural forest is in a slightly different setting in tropical semi-deciduous rain forest where 95 % of all roots from *E. cylindricum* where distributed to depths of between 2.58 and 5.64 m (Freycon et al., 2015).

### 2.5 Evaluation data

As a first step, we evaluate the relevance of the model with respect to real world data (water table depths measurements, soil

moisture, streamflow records, evapotranspiration and gravity data). This section describes the validation data obtained from the Nalohou catchment, which has a comparable size and structure to the one of our conceptual physical model (Fig. 1.a). The Nalohou catchment comprises a small inland valley and is considered to be representative of a cultivated site. It has been the focus of previous studies (Descloitres et al., 2011; Guyot et al., 2012; Hector et al., 2015; Richard et al., 2013). Land surface elevation ranges from 436 to 460 m.a.s.l., and topographic slopes are gentle (mean N-S and E-W slopes correspond to those

imposed on the virtual V-shaped catchment).

In this study, we attempt to reproduce the water table at the inland valley location (P1 in Fig. 1.a), where there is a shallow, 1 m deep, piezometer which aims at documenting the seasonal perched water table, and a deeper (10 m) piezometer monitoring the permanent water table. We also investigate the ability of some model configurations ('saprolite' conditions in the H3 layer, see below) to reproduce the typical water table behavior observed in upstream locations (P2 in Fig. 1.a).

Streamflow measurements have been described in Hector et al. (2015). A Parshall flume was built in 2004 at the outlet of the inland valley (Q2 in Fig. 1.a) and recorded any level change above $10^{-3}$ m, using a level probe (*OTT Thalimedes*).

Two eddy covariance flux towers have been installed respectively on the Nalohou site (where the Flux tower footprint is representative of cultivated, herbaceous-like areas), and on a woodland site 13 km away (Mamadou et al., 2016). Furthermore, an infrared large-aperture scintillometer (LAS) has been deployed from 2006 to 2008 on the Nalohou site (Guyot et al., 2009,

2012). LAS measurements give access to an average sensible heat flux over the footprint area and, by closing the energy budget, this allows to derive the average latent heat flux (i.e. evapotranspiration). In the Nalohou site, the beam path was 2.4 km long, and comprised much more trees than in the flux tower footprint area.

Time-Domain Reflectometry (TDR) probes have been set up on the East bank of the Nalohou catchment at about 150 m eastward of latitude point Y=300 m of Fig. 1.a. Soil moisture content is derived from the measurements at a hourly time step.

Gravity instrumentation originate from the GHYRAF (Gravity and Hydrology in Africa) project, which sought at studying gravity to water storage changes (WSC) relationships (Hinderer et al., 2009, 2012). Gravity instruments were an absolute (AG hereafter) FG5 gravimeter (Hector et al., 2013), a relative superconducting gravimeter (Hector et al., 2014, SG hereafter) and





a CG5 relative portable spring gravimeter (Hector et al., 2015). A specific software, pyGrav, was implemented to process the large CG5 dataset (Hector and Hinderer, 2016). Relative spring microgravimeters measure differences in gravity between stations while other instruments such as SG or AG measure changes at a specific site. Hybrid gravimetry is the joint use of gravity surveys using relative microgravimeters and monitoring of temporal changes at a base station, using an SG and/or AG

(Hinderer et al., 2016). From the hybrid gravity dataset of Nalohou, Hector et al. (2015) produced a map of the seasonal amplitude of gravity changes (their Fig. 11). In this study, we convert gravity changes to water storage changes using a simple scalar admittance (gravity changes to WSC ratio) of 0.04 µGal/mm, in accordance with local topography, and following the admittance calculated by Hector et al. (2015).

## 2.6 Analysis tools

Water storage changes (WSC) analysis is conducted using Empirical Orthogonal Function (EOF) decomposition. EOF decompositions can be seen as principal component analysis applied to the time series of a spatial field. The method produces spatial patterns of variability (in terms of variance), their associated time variation together with a measure of the weight (importance) of each pattern (in terms of explained variance). In other words, the whole data set of coordinates (x,y,t) is decomposed into the weighted sum of modes, where each mode is the product of an EOF spatial pattern (an (x,y) map) with a

normalized time series. The weights are given in terms of "how much of the total signal variance is contained in the associated mode". This method has been used on WSC from hybrid gravity data and neutron probe data at the Nalohou catchment size by Hector et al. (2015) who showed that the first mode could be interpreted as the seasonal signal.

In this study, we make use of the Kling-Gupta efficiency (KGE) criterion (Gupta et al., 2009):

$$KGE = 1 - \sqrt{(r-1)^2 + (\alpha-1)^2 + (\beta-1)^2},$$ (1)

Where,

$$\alpha = \frac{\sigma_S}{\sigma_O}; \ \beta = \frac{\mu_S}{\mu_O},$$ (2)

With r the linear correlation coefficient between simulated and observed time series, and $(\mu_S, \sigma_S)$ and $(\mu_O, \sigma_O)$ are the mean

and standard deviation of respectively simulated and observed time series. KGE has an ideal value at unity.

## 2.7 Analysis Overview of the three virtual experiments

We model an elementary headwater catchment and test its sensitivity to 1) the presence of a low permeability layer underneath an inland valley, 2) the vegetation distribution, and 3) the hydrodynamic properties of contributive areas. For doing so, we build a set of virtual experiments (e.g. Weiler and McDonnell (2004)) that are compared to a reference case (Table 1, the grey



shading represents the reference case). The reference case represents satisfyingly the observed hydrological variables at the Nalohou catchment, as shown in the next section. It is the closest model that matches the authors understanding of the Nalohou system. It includes an inland valley characterized by an impervious layer underlying an upper sandy layer in the valley thalweg, an anthropized vegetation distribution (mostly herbaceous but also trees in the riparian areas), and a clay-like contributive area

(H3 outside the inland valley cells). The three types of virtual experiments are detailed in the following.

The first experiment (first line of Table 4) explores the impacts of the low permeability layer in the valley thalweg, typical of inland valleys, on the hydrological functioning of the catchment. This means that we either model the impervious layer below the H2 horizon (upper part of H3 as shown in Fig. 1.b) in the valley thalweg, or not. Other parameters (hydrodynamic, vegetation…) are kept identical. Note that the reference case includes this impervious layer.

In the vegetation experiments (second line of Table 4), the catchment is either fully tree covered, which is considered as the natural state, or under human influence. Under this so-called anthropization, we consider two different cases: 1) either the catchment is herbaceous-dominated, but with trees kept in the riparian and inland valley fringes areas (as in Fig. 1.a) – this is the reference case, or 2) the catchment is fully herbaceous covered (no trees). First order elementary catchments usually drain into a second order channel, therefore the downstream trees actually represent the riparian area of the second order stream,

here flowing along the X direction.

Table 4

The contributive area drives the subsurface and groundwater transfer toward the inland valley. In the third experiment (third line of Table 3), we test three distinct configurations for the H3 layer outside the valley thalweg, and keeping H1 and H2 properties identical. The configurations are based on two distinct soil profiles commonly found in the Sudanian landscape: a

clayey H3 or a sandy to loamy saprolite H3 layer, both above the hardrock considered as impervious. The reference case considers clay material only in the H3 layer, and the two other cases consider respectively 1) a sandy to loamy saprolite in the H3 layer and 2) a clay zone on the Western side of the virtual catchment and a sandy zone on the eastern side (asymmetric case).

**3 Results**

**3.1 Reference case**

**3.1.1 Water storage changes**

Water storage changes (WSCs) are shown on Fig. 3 as the spatial and temporal patterns of the first mode of the EOF decomposition. While the left panel shows the simulated WSCs, the right panel shows WSCs obtained from the hybrid gravity data by interpolation from point-based values (blue dots). The first mode of the simulated WSCs represents more than 90 %

of the variance. This means that more than 90 % of the total signal variance is contained in the first EOF spatial pattern multiplied by its associated time series (Fig. 3, lower panel, red line). The time series shows that this mode corresponds to the



seasonal cycle (the common part of the signal which explains most of the total variance). In that sense, we can directly interpret the spatial patterns (i.e. the EOFs) as maps of seasonal water storage amplitudes. The first mode of gravity-derived WSCs also explains a significant part (79 %) of the total variance. As this map derives from an interpolated field, and although gravity data is spatially integrative, observed geometries should not be over-interpreted.

5                                                    Figure 3

The reference simulation is built largely following our understanding of the Nalohou catchment. Particularly, it does include an inland valley (a low permeability layer in the upper part of the H3 horizon) and the clayey material in the H3 layer outside the inland valley corresponds to what is found in the West bank of the Nalohou catchment. To this regard, the simulated spatial pattern matches two important features of the observations: a high amplitude in the inland valley and a low amplitude on the

clayey, West bank, area. The high simulated amplitude in the downstream section (500 to 700 m) is due to tree withdrawals during the dry season. Gravity data do not cover this part of the catchment which prevents the comparison with the simulated data. The low amplitude on the sides of the catchment is due to the impermeable clay in the H3 horizon, which prevents water to percolate downward and favors lateral transfer to the valley thalweg where water accumulates during the wet season, and drains down toward the riparian areas during the dry season. High amplitude is simulated along the whole valley thalweg, and

not only in the inland valley as the observation show.

### 3.1.2 Water table

Fig. 4.a shows simulated time series of saturation profiles at P1, in the inland valley for the reference simulation. Red lines show the simulated water tables, while white lines show the observed water tables. The model is able to reproduce seasonal fluctuations of both the permanent water table and the temporary perched water table. Although numerous discrepancies exist

(e.g. the seasonal amplitudes of the permanent water table), mainly because of the horizontal and vertical variability of hydrodynamic parameters which have been considered constant in layer H3, the first order of this complex behavior is reproduced. Particularly, the interannual behavior of the permanent water table is rather satisfying, despite a slight phase shift, but also higher frequency responses of the shallow perched water table at the core of the rainy season are well reproduced.

In the virtual experiments later shown, the saprolite case (sandy H3 layer everywhere but in the inland valley) is considered,

as it is a common feature of the environment, particularly in the East bank of the Nalohou catchment. Fig. 4.b shows simulated time series of saturation profiles at P2, in the upstream area for the saprolite simulation. The simulated permanent water table matches satisfyingly the observation representative of this sandy H3 unit.

Figure 4

### 3.1.3 Streamflow

Fig. 5 shows simulated and observed streamflow expressed in mm/h, which allow the comparison between the Nalohou catchment (0.16 km²) and the V-shape model (0.216 km²). We chose to focus the plot on the baseflow part of the time series, cutting the highest flow peaks, for two reasons: 1)  Hector et al. (2015), noted that in such settings baseflow is the major



contributor to total measured streamflow (about 80%) and 2) the following sections will present semilog plots of the different experiments simulated, which allow to display the whole range of streamflow values, but with less intuitive interpretation. Overall, the match is satisfying, and the simulation respects baseflow onset, and recession behaviours. The only negative KGE value for the year 2009 is composed by ($r = 0.96, \alpha = 1.56, \beta = 1.84$), and is therefore mostly explained by a high bias and too high simulated variance, while the time phasing matches well. It has to be noted that the simulation produces higher flow peaks than observed for some events, but not all (see further Fig. 9). Discrepancies between observed and simulated streamflow are higher in drier years (i.e. 2011). This is due to the complexity of connectivity between different contributive compartments, and for which our idealized simulation probably still lacks some significant components. The simulated diurnal cycle in the ponded water (from which streamflow is calculated) is a model artifact due to the ponded cell size (20 m wide here).

Figure 5

### 3.1.4 Evapotranspiration

Fig. 6.a shows the LAI and SAI forcing time series used in this study for the two vegetation covers. Fig. 6.b (resp. 6.d) shows the simulated daily evapotranspiration spatially averaged over herbaceous-covered cells out of the thalweg in light blue line for the reference simulation (resp. the saprolite simulation). The shaded area gives the whole spatial range of daily values between minimal (lower black line) and maximal (upper black line) spatial evapotranspiration. Fig. 6.c (resp. 6.e) is similar, but for tree-covered cells. Reference (clay in the H3 unit everywhere but below the inland valley) and saprolite simulations are compared because the Nalohou Flux tower is located on a saprolite-like material. The saprolite simulation actually matches more closely the observed evapotranspiration in the dry season.

During the dry season, trees renew their leaves and show a low LAI and limited evapotranspiration. The herbaceous have been burnt, and only some scattered shrubs and trees imply a non-zero LAI and little evapotranspiration. During the transition period between dry and wet seasons, trees LAI increase, herbaceous grow and evapotranspiration increase with both soil moisture (not shown) and plants ability to transpire with a higher LAI. During the wet season, the evapotranspiration is never water limited (Mamadou et al., 2016). During the transition period between wet and dry seasons, the herbaceous layer dries up, and LAI is being replaced by SAI which diminishes drastically the evapotranspiration. Herbaceous are eventually burnt at the end of this transition period, while trees start renewing their leaves.

Overall, the spatially averaged simulated evapotranspiration matches the data, over both herbaceous and tree covers. At the end of the wet season, the simulated evapotranspiration (dominated by transpiration) is a little higher than the EC observations, but not higher than the LAS observations in 2008. EC is known to underestimate evapotranspiration (Foken, 2008).

Figure 6

### 3.1.5 Soil moisture

Because soil moisture probes are located on the East side of the Nalohou catchment, and the reference simulation is built based on our understanding of the West side (hence with H3 properties different to those at the physical location of the soil moisture





probes), observed soil moisture are here compared to the 'saprolite' simulation where H3 outside the inland valley is composed by sandy-like saprolite (Fig. 7). Simulations match the observed temporal patterns in a satisfying way, including recession behaviors.

Figure 7

**3.1.6 Annual water budget**

The water budget simulated for the reference case is shown in Table 5. As expected, evapotranspiration dominates the water budget (from 58 to 83 % of yearly precipitation amount) but shows a marked interannual variability as simulated in the area at basin scale (Cornelissen et al., 2013; Séguis et al., 2011) and from a review on different other spatial scales (Hector, 2014). For dry years, the ET part (resp. Q part) of the budget is higher (resp. lower) than for wet years. Interannual water storage changes are rather limited and do not seem to exhibit memory effects. In other words, the relative part of (ET+Q) in the water budget is stable over years. If we now consider the absolute values of the water balance terms, the difference between a wet year (2010) and the following dry year (2011) is 593 mm in rainfall, 135 mm in evapotranspiration, 420 mm in streamflow and 38 mm in water storage. Streamflow is the most affected term in a dry year: it undergoes a drastic loss of water, about 3 times greater than that of evaporanspiration, almost equal to its interannual average (454 mm). It results that the interannual standard deviation of Q is more than twice as large as the standard deviation of ET: the interannual variability in precipitation (standard deviation = 225 mm) is mostly compensated by streamflow (151 mm) and to a lesser extent by evapotranspiration (64 mm) and water storage changes (26 mm).

Table 5

**3.2 Virtual experiments**

**3.2.1 Annual water budgets**

The most important variations in yearly water budgets occur when comparing a tree-covered catchment versus an herbaceous-covered catchment (Table 6). Mean interannual evapotranspiration is higher (resp. lower) and streamflow is lower (resp. higher) for the trees (resp. herbaceous) -covered catchment, while the reference case shows a somewhat intermediate behavior. Interannual variability (i.e. as shown by the standard deviation) shows the opposite behavior: it is lower (resp. higher) for the trees than for the herbaceous. Unlike other cases, interannual water storage changes may become significant in the tree-covered experiment, with a net decrease in water storage during the dry years. It results in a higher interannual standard deviation of the storage term.

The presence of the impermeable layer (in the reference case) has very little effect on the water budget, lowering the streamflow of 2 % (9 mm) on average, in favor of evapotranspiration, with respect to the case without the impermeable layer.

The reference, clayey, contributive area evapotranspirates slightly more than the saprolite-like contributive area, of 5% on average, while the mixed simulation (clayey left bank, saprolite right bank) shows an intermediate behavior.



The impact of the different simulations on the baseflow to streamflow ratios is very limited. The absence of the impermeable layer increases the baseflow of less than five points depending on years. For wet years (i.e. 2010), there is relatively less baseflow when there is an impermeable layer. The vegetation seems not to influence the baseflow to streamflow ratio, while a more permeable (saprolite-like) contributive area exhibit slightly more baseflow (less than five points) than a low-permeability, clayey contributive area.

Table 6

### 3.2.2 Water storage

Fig. 8 shows water storage changes as the first modes of the Empirical Orthogonal Function (EOF) decompositions for all tested cases. In all cases, the first mode represents more than 90 % of the variance.

The most striking discrepancies arise when comparing the results related to the contributive area experiments (Fig. 8.a,e,f). While the clayey reference case (Fig. 8.a) shows low seasonal amplitude in the valley banks and higher amplitude along the valley axis including the inland valley, the saprolite-like contributive areas (Fig. 8.e) show rather opposed behavior with higher amplitude on the banks and lower amplitude in the valley axis. In the reference case, the southern, upstream edge of the inland valley exhibit a break in the seasonal amplitude, while this is not visible in the saprolite case (in this case there is no specific imprint of the inland valley). In the mixed case (Fig. 8.f), the southern, upstream edge of the inland valley also exhibits a different seasonal amplitude, but of opposite behavior as in the reference case.

In the vegetation experiment (Fig. 8.a,c,d), the tree-covered catchment (Fig. 8.c) shows higher seasonal amplitude than the herbaceous-covered catchment (Fig. 8.d), while the reference case shows a closer pattern to the herbaceous-covered catchment. In the time series, seasonal WSC increase occurs earlier on herbaceous cover than on trees.

The presence of the impermeable layer in the inland valley diminishes the seasonal amplitude of the thalweg (Fig. 8.b).

In all cases, there is low seasonal amplitude at the catchment outlet (the northern blue patch). This is the imprint of the topography (all cells converge to that point) and the model boundary condition: the only way for the water to exit the domain, apart by evapotranspiration, is through this outlet.

The mixed simulation characteristics are actually the closest to the conceptual model presented by Hector et al. (2015). And the observed spatial pattern (Fig. 2) matches several features of the mixed simulation: a high amplitude in the inland valley, a low amplitude on the clayey, left bank, area, and some intermediate amplitude in the saprolite-like right bank area. While this is satisfying, it has to be noted that the descriptive model proposed by Hector et al. (2015) lacks some details, because in this mixed simulation (which follow their descriptive model), while streamflow, evapotranspiration and spatial pattern of WSC are well reproduced, the simulated permanent water table completely fills up the inland valley due to high permeability in the H3 layer on the right bank (see further). There must hence be some barrier to the lateral flow feeding the inland valley. However, it would be interesting to investigate the existence of other inland valleys which do not show such disconnexion between the upper perched water table and lower permanent water table. Our study would anticipate that, all scaling issues discarded, such an inland valley would produce a similar water budget.



Figure 8

### 3.2.3 Streamflow

Yearly simulated streamflow time series (sampled on cell Q1, Fig. 1) are shown on Fig. 9 for the year 2011 and as semilog plots, with the reference time series repeated in all plots in black. Overall, streamflow simulations are rather similar. Much of the differences among the cases are related to the first part of the rainy season, in the timing of the streamflow onset and the streamflow amplitude during this period (this is also true for other years, not shown). Differences in the dry season are very low in terms of water amount (remind the log scale), but exhibit clearly different behaviors: some cases are able to sustain a low baseflow during the dry season, while some others including the reference case prevent water flow during the dry season. The presence of the impermeable layer in the inland valley (reference case) induces a higher streamflow in August (Fig. 9, upper panel). On the other hand the absence of the impermeable layer favors infiltration and maintains a low baseflow longer in the dry season, fed by the permanent water table which is then connected all across the catchment.

The tree-covered catchment (Fig. 9 middle panel) tends to exhibit lower streamflow in August. The storm peaks are much attenuated, as compared to the reference case. The herbaceous-covered catchment shows a higher streamflow in the dry season and the early wet season, with slightly higher storm peaks than in the reference case. One may note how the reference Q follows the trees Q in the early season before August and then migrates to meet the herbaceous Q that it follows until the reverse migration is observed in November. This likely denotes the course of the contributive area extension through the years, trees being only in the downstream section in the reference case.

The simulation with saprolite-like contributive area (Fig. 9 lower panel) shows a higher dry season and early wet season streamflow as compared to the reference case and the mixed case. At the core of the rainy season, in august-september, the saprolite simulation shows a lower streamflow.

Figure 9

### 3.2.4 Water table

Fig. 10 shows the simulated water tables at two locations, in the inland valley (P1, Fig. 1), and in the upstream contributive area (P2, Fig. 1). When present, perched water tables are shown too.

As expected, the presence of the impermeable layer (Fig. 10.a, as in the reference case) creates two water tables: a deep permanent, low amplitude, water table, and a shallow seasonal water table within the sandy layer of the inland valley (see Fig. 4). This is not seen when the impervious layer is removed (no I. valley experiment), and where only a strong amplitude permanent water table is present. On the upstream contributive area (Fig. 10.d), no difference is observed.

All vegetation experiments (Fig. 10.b) simulate perched water table in the inland valley. The tree-covered catchment shows a deeper water table than the herbaceous-covered catchment, while the reference simulation shows an intermediate behavior. On the upstream contributive area (Fig. 10.e), the herbaceous cover matches the reference simulation. The strong water table



drawdown induced by tree water uptake lowers the water table down within the clay layer, and no more seasonal variations are observed. In fact, the simulation did not even reach an equilibrium state in that point.

When the whole catchment or half of the catchment deeper layer (H3) is saprolite (Fig. 10.c), the fissured basement below the inland valley (in H3) fills up and no perched water table is observed. On the upstream contributive area (Fig. 10.f), however,

there is a strong seasonal water table amplitude (as also shown in Fig. 4), not observed in the clayey, reference simulation.

Figure 10

## 4. Discussion

### 4.1 Inland valley

The presence of the impermeable layer in the inland valley does not influence much the yearly water budget (1 % -resp. 2 %-

change in the interannual mean for ET -resp. Q-), although the water table behavior in the valley thalweg (Fig. 10.a) and the spatial pattern of seasonal water storage changes are much different (Fig. 8.a,b). The impervious layer prevents vertical infiltration of water below the inland valley, which participates in the creation of a perched water table. Without the impervious layer, the permanent water table would fill up during the rainy season until it would reach the depth of the perched layer of the inland valley (as simulated with an impermeable layer) and act similarly. The period when this permanent water table would

reach the level of the perched water table (of the reference case) would correspond to the period of matching streamflow in Fig. 9 (upper panel). Despite little differences in terms of annual water budget, the sustained low baseflow during the dry season in the absence of the inland valley, could have significant eco-hydrological consequences downstream. This should be investigated at the scale of larger hydrosystems (as the Upper-Ouémé catchment).

In similar context, Giertz and Diekkrüger (2003) observed significantly higher total runoff amount, peak flow and a longer

runoff period in a catchment containing a larger inland valley area (catchment size: 3.5 km², inland valley area: 0.19 km²), than in a nearby catchment area (catchment size: 3.2 km², inland valley area: 0.07 km²), both located in the Sudanian area in northern Benin. Although we do not observe significant differences in the water budget, we do observe longer runoff period. The catchment size may play a role in the differences with the observations from Giertz and Diekkrüger (2003), and this could be further investigated. One has to note that the inland valley this study is based on is in the low range of inland valley size

distribution in the area, following a survey conducted on 817 inland valleys (Giertz et al., 2012). According to this study, 18 % of inland valleys of the area have surface areas lower than 1 ha (and 80 % lower than 10 ha), as the inland valley comprised in the Nalohou catchment (0.3 ha).

### 4.2 Vegetation distribution

The vegetation distribution does influence the yearly water budget. For evapotranspiration, for instance, average yearly amount

vary between 74 % and 62 % of the precipitation for respectively a fully tree-covered and an herbaceous-covered catchment (68 % for the reference case). The difference in average yearly is 16 % (with respect to the trees simulation) with larger





difference for dry years (24 % for 2011) than for wettest (12 % for 2010). More important, the interannual variability of evapotranspiration (i.e. as shown by the standard deviation) is strongly reduced (26 %) for a tree-covered catchment (53 mm) as compared to an herbaceous-covered catchment (72 mm).

Evapotranspiration is higher for the tree-covered catchment, mostly because trees are able to get water from the ground for a
longer period than herbaceous cover during the dry season. This is because trees have a longer non-zero LAI period, as they take a short time to renew their leaves and are not burnt by farmers as the herbaceous. They also have access to deeper water thanks to the deeper root system. Finally, trees undergo higher interception and subsequent evaporation thanks to their low angle leaf orientation and higher LAI and SAI. Between a herbaceous covered catchment and a tree-covered catchment, yearly average interception losses increases from 3.2 % to 6.5 % of annual precipitation. Herbaceous dry up at the end of the rainy
season, and are further burnt by local farmers. When transpiration ceases, evaporation is also limited due to the lack of shallow soil water.

The consequence of a higher evapotranspiration for the tree-covered catchment is a low yearly streamflow budget of 28 % on average as compared to a herbaceous-covered catchment, ranging from 20 % for the wettest year (2010) to 47 % for a dry year (2011). This occurs because during the dry season, when there is no or negligible streamflow, the trees lower the total water
storage of the catchment, as shown by the lower simulated water tables (Fig. 10.b,e) and the higher seasonal water storage amplitudes (Fig. 8.a,c). In the early period of the rainy season in the tree-covered simulation, rainwater takes a longer time to infiltrate and sufficiently recharge the water table until streamflow finally reaches the streamflow level of the herbaceous-covered simulation (Fig. 9. Middle panel). This is particularly visible in dry years (i.e. 2011), where highly intermittent rainfall and long dry periods during the rainy season prevent significant percolation and further delay the "saturated" condition of the
catchment.

These results must be tempered by the fact that we used a highly idealized "herbaceous cover", which is actually a mixture of different vegetation classes (perennial shrubs, isolated trees and perennial and annual grass species, mainly *graminae*) and "trees" which do not take into account differences among species. For instance, in Sudanian area, a recent study showed an example where the woody cover of agroforestry systems transpires less than forest cover, not only due to lower tree density
but also to species composition that are selected in agroforestry practices (Awessou et al., 2016).

As a consequence, tree covers buffer the hydrological cycle by 1) lowering streamflow, 2) reducing interannual variability of evapotranspiration, 3) increasing interanual water storage change by decreasing the water storage during the dry season and 4) increasing interception losses. In the Sudanian area, streamflow has decreased since the 80's, following precipitation decreases, sometimes twice as much (Descroix et al., 2009). LULC changes occurred simultaneously, mostly by conversion
of savannah or woodlands to crops (e.g. Eva et al. (2006)). The question of the attribution of the hydrological changes to either climate forcings or LULC changes has not yet been solved. While it is likely that LULC changes played a minor role since the 80's (because of the observed streamflow decrease), if the still ongoing deforestation maintains its pace, there is a chance that the hydrological cycle could be significantly altered in the future due to LULC changes. For instance, Yira et al. (2016) used




a process-based hydrological model to show that converting savannah to crops in the last 25 years in a sudanian catchment in Burkina Faso led to Q increase and ET decrease, a result consistent with what we can anticipate from our study.

### 4.3 contributive areas

When sand-like transmissive hydrodynamic properties are set for lateral and upstream areas of the inland valley (the saprolite case, as opposed to the clay-like reference case), the simulation allows the permanent water table to fill up the area below the inland valley (Fig. 10.c). This means that simulated seasonal water storage amplitudes are lower in the valley thalweg than in the valley sides (Fig. 8.e), showing the opposite pattern of the reference case, where water storage amplitudes are stronger in the valley thalweg (Fig. 8.a). Despite these strong spatial differences, the annual water budget exhibit little differences.

In terms of streamflow generation, the subsurface behavior of the inland valley is rather similar in the saprolite case and the reference case, as whether the shallow perched water table is connected or not to the deeper permanent water table does not matter in its ability to supply the streamflow. In both cases, there is a sufficiently high permeability layer (H2 Hardpan layer in the reference simulation, and H3 in the saprolite-like simulation) which allows lateral transfer to feed the inland valley. However, although the annual water budgets show little differences, there may still be significant differences in temporal

distribution of fluxes. For instance, the saprolite case sustains a higher baseflow (Fig. 9 lower panel). Also, the simulated evaporation differences during the dry season, over herbaceous cover (see Fig. 6.d) imply different surface feedbacks on the atmosphere, which may control larger scale atmospheric features. Evapotranspiration is known to be linked to soil properties through soil moisture availability. Descloitres et al. (2011) showed in the area, for instance, that evapotranspiration as measured by a large aperture scintillometer was controlled by soil properties. To check the consistency of these simulated evaporation

differences during the dry season, evapotranspiration should be monitored on a hardpan-clay soil profile, as they are widely present in the region.

### 5. Conclusions

In this paper, we studied the hydrological functioning of Sudanian inland valleys and their sensitivity to land cover and contributive areas through deterministic sensitivity experiments using a physically-based critical zone (CZ) model applied on

a synthetic catchment which comprises an inland valley. This is a first approach to try to investigate what can control and explain the behavior of an inland valley.

We first showed that a CZ model could be used to reproduce the complexity of a small hydrosystem (0.16 km²) as an inland valley and most of observed water fluxes and temporal dynamics as streamflow, evapotranspiration, soil moisture, water table levels, or water storage. Particularly, we confirmed that the interannual variability in precipitation (standard deviation = 225

mm) is mostly compensated by streamflow (151 mm) and to a lesser extent by evapotranspiration (64 mm) and water storage changes (26 mm).



We found that yearly water budgets were almost insensitive to the presence (or absence) of the low permeability layer commonly found below inland valley, although intra-seasonal variations in water storage and water table showed significant differences. Early season streamflow was affected by a slight delay when the low permeability layer of the inland valley was virtually removed, which corresponds to the time required for water to percolate below the inland valley. Higher water table

levels allowed by the enhanced infiltration below the inland valley was then simulated, which could sustain a low but permanent baseflow during the dry season, which could have significant impact downstream. This virtual experiment confirms that the observed streamflow dynamics, particularly in the dry season, is partly due to the low permeability layer, a typical inland valley feature due to the accumulation of clays (Blavet, 1997; Brabant, 1991; Von Der Heyden, 2004).

On the other hand, yearly water budgets were much more sensitive to the vegetation distribution: average yearly ET for a tree-

covered catchment (944 mm) exceeds that of herbaceous-cover (791 mm). ET differences between the two covers vary between 12 and 24 % of the precipitation of the year for the wettest and driest year, respectively. Therefore, the tree-covered catchment produces a yearly streamflow budget 28 % lower on average as compared to a herbaceous-covered catchment, ranging from 20 % for the wettest year to 47 % for a dry year. Trees also buffer interannual variability in ET by 26 %. Extensive LULC changes in the last thirty years, resulting in forest and woodland conversion to agricultural lands, still undergoing today, calls

for paying a closer attention to the distinct roles of climate changes and land cover changes on the resulting hydrological functioning of Sudanian systems.

We also found little sensitivity of the yearly water budget to the hydrodynamic properties of upstream and lateral areas, although these implied completely opposite behaviors in the intraseasonal space-time patterns of water storages and significant changes in water table depths together with different dynamics and amplitudes of fluxes (e.g. streamflow, soil evaporation).

Higher water table levels in the valley thalweg, allowed when materials that are more permeable are taken into account, foster a low but permanent dry season baseflow.

To summarize, it is shown that while vegetation cover significantly change water budgets, pedological features have much more impacts on intraseasonal hydrological processes (sustained/unsustained baseflow in the dry season, catchment water storage redistribution…) than on yearly water budgets amounts, and therefore potentially significant impacts on water-

dependent ecosystems, but also water uses as hydropower generation.

This study was limited to an elementary headwater catchment, and results derived here may not apply at larger scales like basins with larger inland valley/catchment area ratios, or different soil profile distributions. Larger scales models should be run to investigate these effects, and thus large scale spatial features should be mapped, which poses significant issues regarding subsurface structure. Indeed the in situ measurements of the underground structure are missing and the structure is not directly

available through remote sensing products for instance.



## 6. Acknowledgements

The authors would like to thank many people who helped in acquiring, maintaining and processing data and instruments in Djougou: S. Tahirou, F. Littel, J.D. Bernard, B. Luck, N. LeMoigne, I. Imorou, E. Pagou, T. Ouani, S. Afouda and M. Wubda. We would also like to thank the project partners who consented to the use of their infrastructures and provided valuable

information and advices: A. Zannou and J.C. Gbodogbé (Direction Générale de l'Eau, Cotonou, Bénin) and N. Yalo (University of Abomey-Calavi, Bénin).

The AMMA-CATCH regional observing system was set up thanks to an incentive funding of the French Ministry of Research that allowed pooling together various pre-existing small-scale observing setups. The continuity and long-term perenity of the measurements are made possible by an undisrupted IRD funding since 1990 and by a continuous CNRS-INSU funding since

2005. AMMA-CATCH also received support from the *LabEx OSUG@2020*.

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





|  | $\theta_S$ | $\theta_r$ | Ks (m/s) | \|hg\| (m) | n |
|---|---|---|---|---|---|
| **H1** | 0.35-0.45 | 0.01-0.05 | $5.8.10^{-6}$-$5.7.10^{-5}$ | 0.2-0.6 | 1.6-2.2 |
| **H2** | 0.35-0.45 | 0.05-0.1 | $1.2.10^{-7}$-$3.5.10^{-5}$ | 0.2-0.6 | 1.5-1.8 |
| **H3 (Clay)** | 0.4-0.5 | 0.05-0.1 | $1.2.10^{-8}$-$1.2.10^{-6}$ | 1-1.5 | 1.15-1.5 |
| **H3 (Saprolite)** | 0.4-0.45 | 0.05-0.1 | $1.2.10^{-6}$-$5.7.10^{-5}$ | 0.7-1.5 | 1.2-2 |
| H3 (Inland Valley) | 0.2-0.35 | 0.01-0.05 | $1.2.10^{-6}$-$5.7.10^{-5}$ | 0.7-1.5 | 1.5-2.2 |

**Table 1: ranges of hydrodynamic properties in the area. From Hector, 2014.**

| Property | Value |
|---|---|
| Extension X | 300m |
| Extension Y | 720m |
| Extension Z | 10.5m |
| DX (grid resolution) | 20m |
| DY | 20m |
| DZ | variable: 0.13 – 1.19m |
| duration | 8yrs |
| Dt | 0.5hr |
| Slope X | 2 % |
| Slope Y | 4 % |

**Table 2: model setup**

|  | $\theta_S$ | $\theta_r$ | Ks (m/s) | \|hg\|(m) | n |
|---|---|---|---|---|---|
| **H1** | 0.4 | 0.01 | $1.11.10^{-4}$ | 0.3 | 1.8 |
| **H2** | 0.36 | 0.01 | $2.8.10^{-4}$ | 0.3 | 1.65 |
| **H3 (Clay)** | 0.45 | 0.056 | $5.6.10^{-9}$ | 1 | 1.3 |
| **H3 (Saprolite)** | 0.3 | 0.004 | $1.10^{-5}$ | 1.3 | 1.76 |
| H3 (Inland valley) | 0.3 | 0.004 | $2.8.10^{-5}$ | 0.33 | 1.65 |

5 **Table 3: hydrodynamic properties for the virtual V-shaped catchment.**

| Inland valley low permeability layer | yes | no | - |
|---|---|---|---|
| Vegetation | anthropized (trees only in riparian area and inland valley fringes) | tree covered | Fully anthropized (no trees) |
| contributive area (H3 outside the valley thalweg) | clayey | sandy to loamy (saprolite) | Asymmetric (clay in the West, Sand in the East) |

**Table 4: virtual experiments. Reference case is shown in grey**

| Year | P(mm) [(%)] | ET (mm) [(%)] | Q (mm) [(%)] | S (mm) [(%)] |
|---|---|---|---|---|
| **2006** | 913 [100] | 775 [85] | 197 [22] | -59 [-6] |
| **2007** | 1214 [100] | 829 [68] | 374 [31] | 11 [1] |
| **2008** | 1211 [100] | 799 [66] | 400 [33] | 12 [1] |
| **2009** | 1496 [100] | 931 [62] | 551 [37] | 14 [1] |
| **2010** | 1527 [100] | 881 [58] | 644 [42] | 3 [0] |
| **2011** | 934 [100] | 746 [80] | 224 [24] | -35 [-4] |
| **2012** | 1423 [100] | 848 [60] | 529 [37] | 45 [3] |
| **Mean ± std (2007-2012)** | 1301 ± 225 | 839 ± 64 | 454 ± 151 | 8 ± 26 |

**Table 5: water budgets for the reference case. Mean and standard deviations are calculated on the period 2007-2012**




|  | ET (mm) | Q (mm) | S (mm) |
|---|---|---|---|
| reference | 839 ± 64 | 454 ± 151 | 8 ± 26 |
| No Inland valley | 830 ± 64 | 463 ± 153 | 8 ± 24 |
| trees | 944 ± 53 | 361 ± 144 | -5 ± 54 |
| herb | 791 ± 72 | 500 ± 151 | 10 ± 17 |
| saprolite | 793 ± 60 | 495 ± 153 | 13 ± 36 |
| mixed | 816 ± 62 | 475 ± 154 | 10 ± 28 |

**Table 6: Interannual mean and standard deviation of simulated evapotranspiration (ET), streamflow (Q) and storage (S) for the different virtual experiments and for the period 2007-2012**

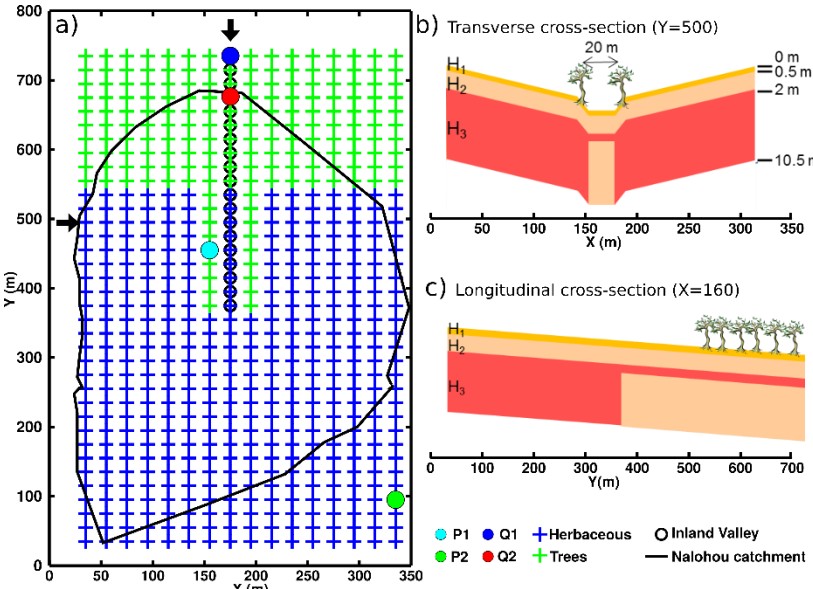

5    **Figure 1 : a) catchment mesh. Blue and green crosses indicate the vegetation distribution for the reference case. Inland valley cells are shown as black circles. P are water table sampling locations and Q are streamflow sampling locations. b-c) domain cross sections schematically made of three soil types: high-permeability sandy soils (yellow), hard-pan, or fissured basement (orange) and low permeability clay accumulation (red).**





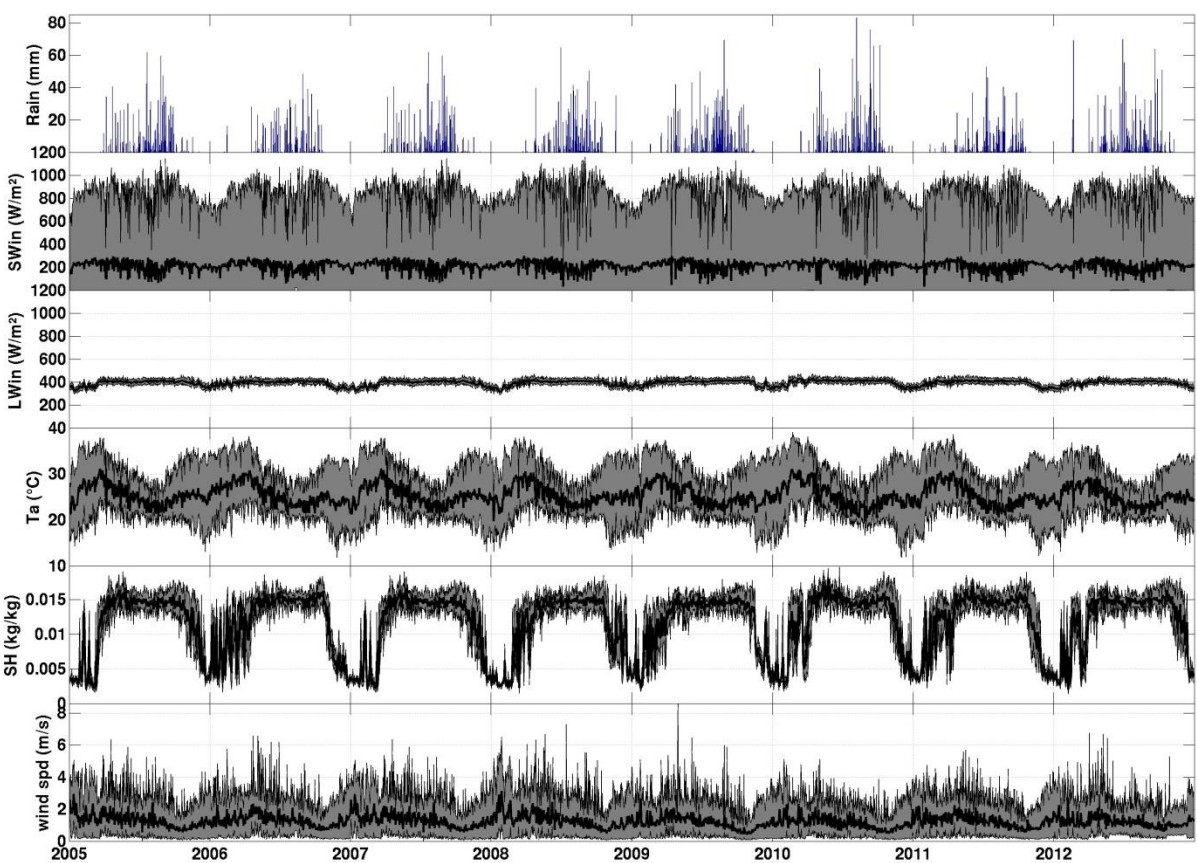

**Figure 2 : Model forcing: upper panel: daily precipitation. SWin = incoming short waves, LWin = incoming long waves, Ta = air temperature, SH = specific humidity, wind spd = wind speed. Data is shown as daily mean (black curve) and daily minima and maxima (shaded area)**



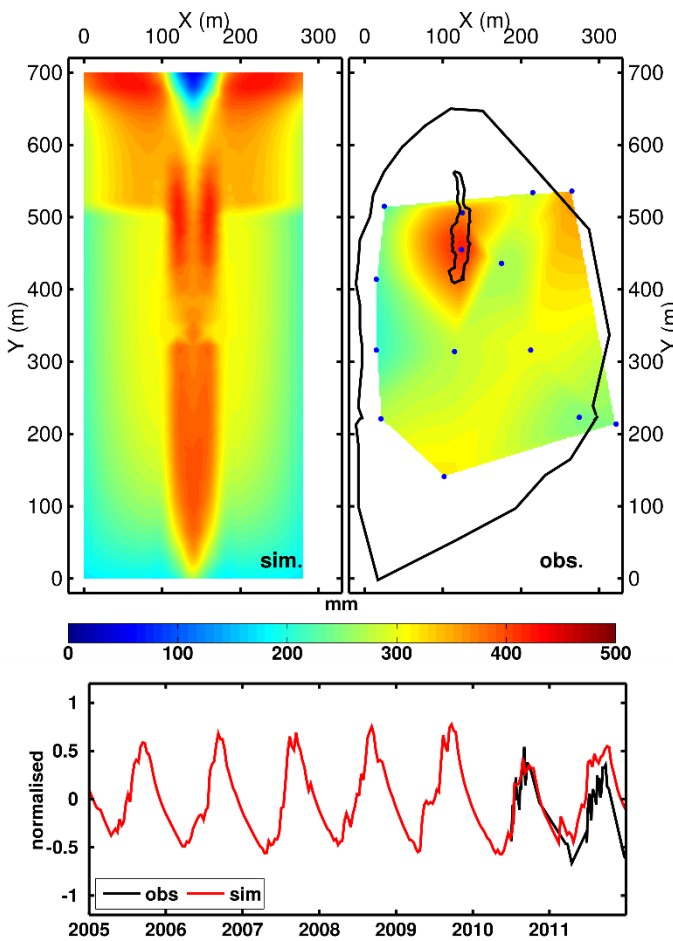

**Figure 3 : Empirical Orthogonal Function decomposition of water storage changes (WSC). Upper panels: spatial patterns of WSC for the 1st mode of EOF. Left: simulated WSC, right: observed, gravity-derived WSC at the Nalohou catchment (blue points are station locations). The outline of the inland valley is shown for the Nalohou catchment. Lower panel:  temporal pattern of the first mode of simulated and observed WSC.**





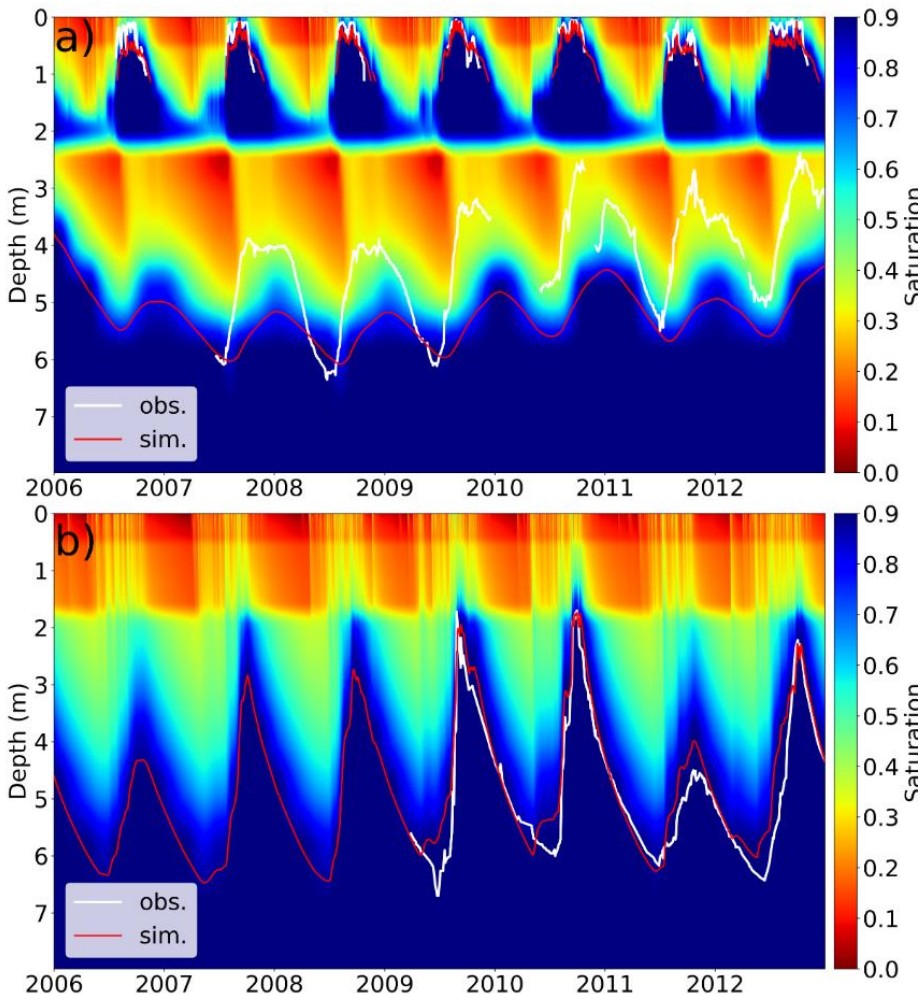

**Figure 4 : simulated saturation, and observed and simulated water table. a) In the inland valley (at P1), and for the reference simulation (clayey H3 layer), b) in the upstream area (at P2) and for the saprolite simulation (sandy H3 layer)**





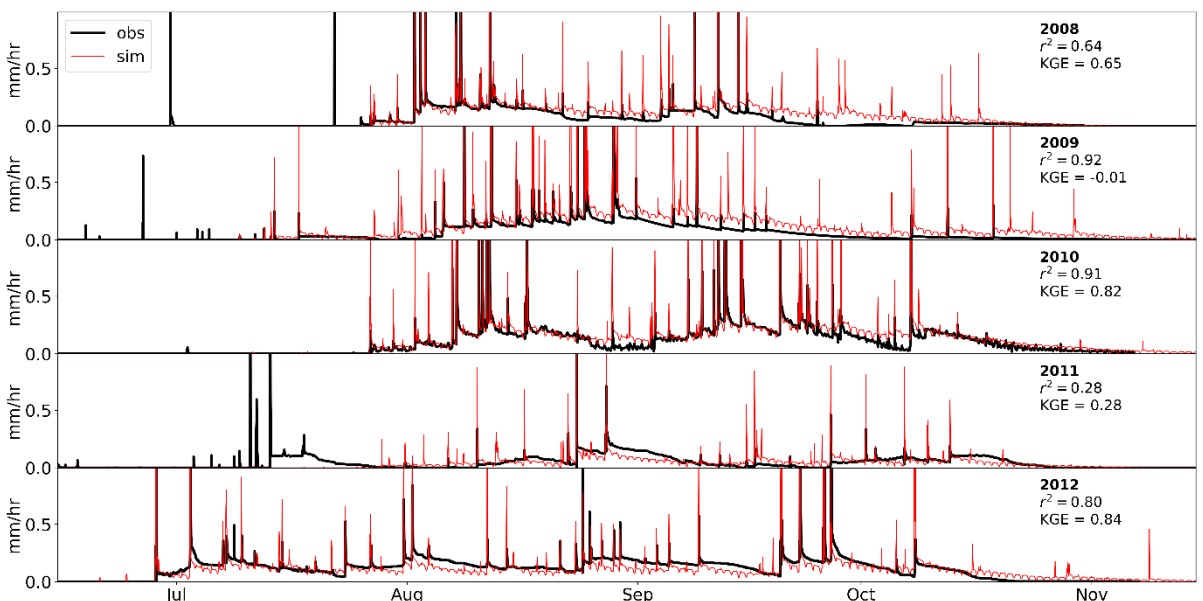

**Figure 5 : Observed and simulated streamflow for the wet seasons of the period 2008 to 2012 at station Q2**

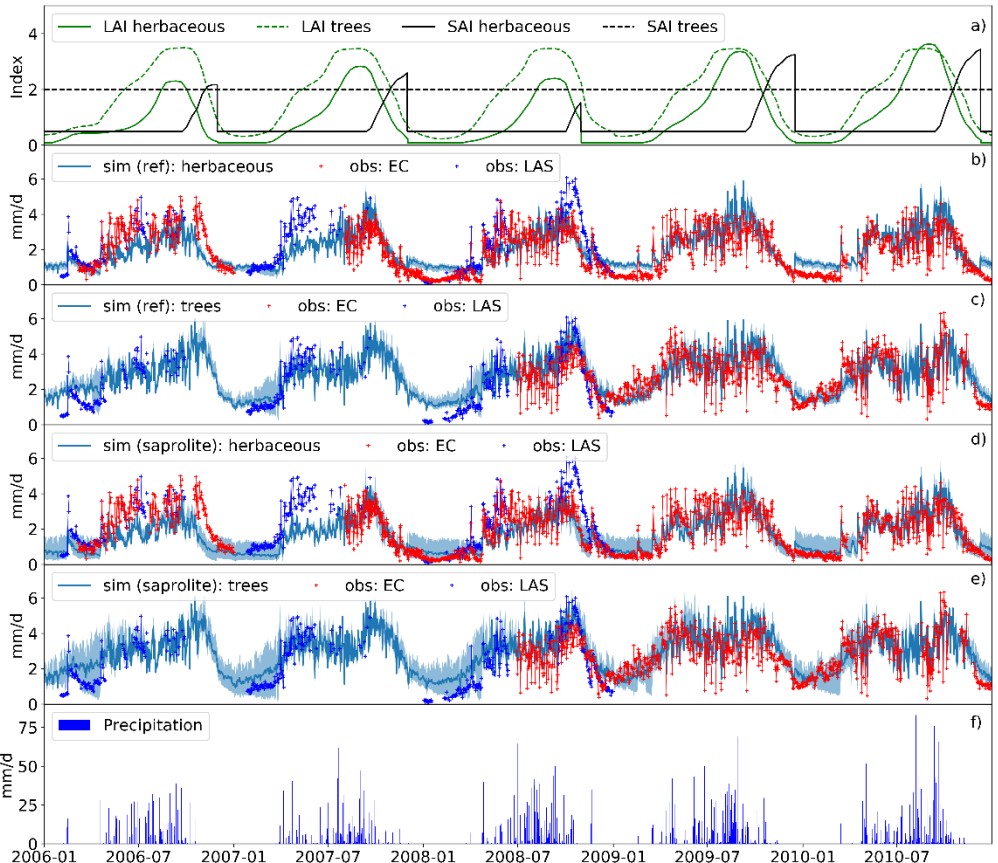





**Figure 6 : a): LAI and SAI forcings for tree and herbaceous covers. b-e): Observed and simulated evapotranspiration over herbaceous cells (b) and trees cells (c) for the reference simulation, and the saprolite simulation (d and e). Blue line shows the spatial average over all cells of the respective cover out of the inland valley, and the shaded area show the extent of spatial maxima and minima. f) precipitation.**

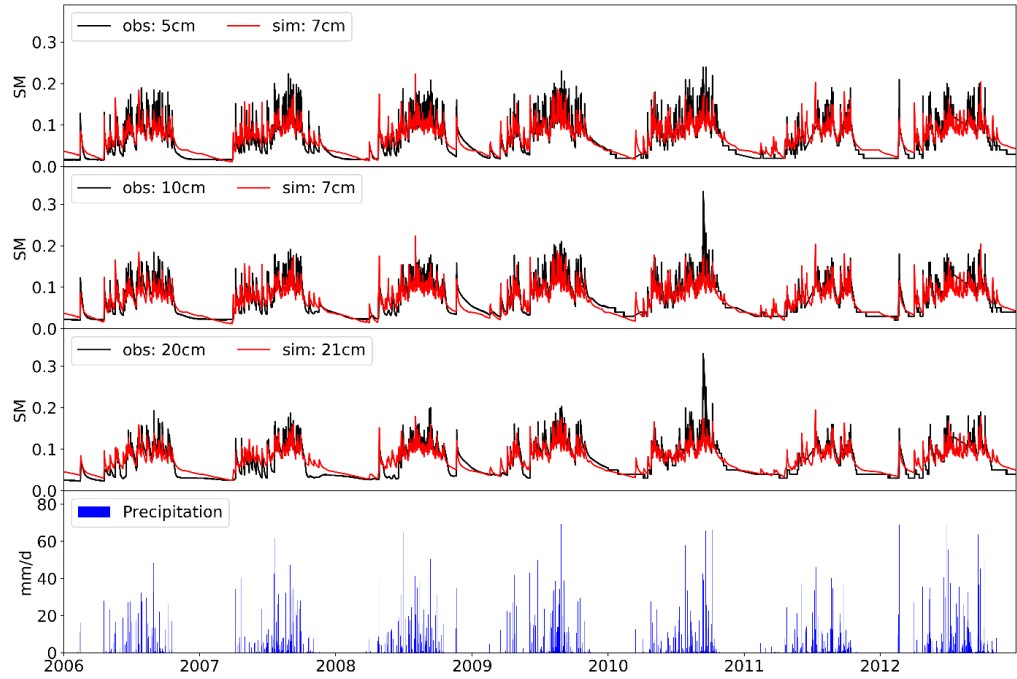

**Figure 7 : observed and simulated soil moisture for the saprolite simulation at location P2**





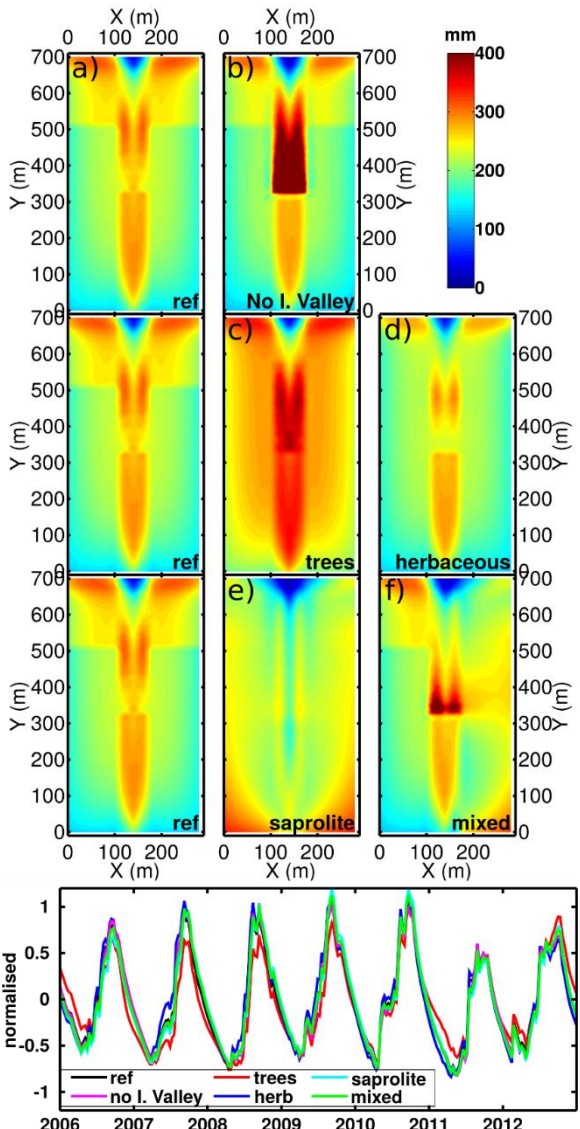

**Figure 8 : Simulated water storage changes : First mode of the EOF for all cases: upper panel: spatial patterns. Lower panel: temporal patterns.**





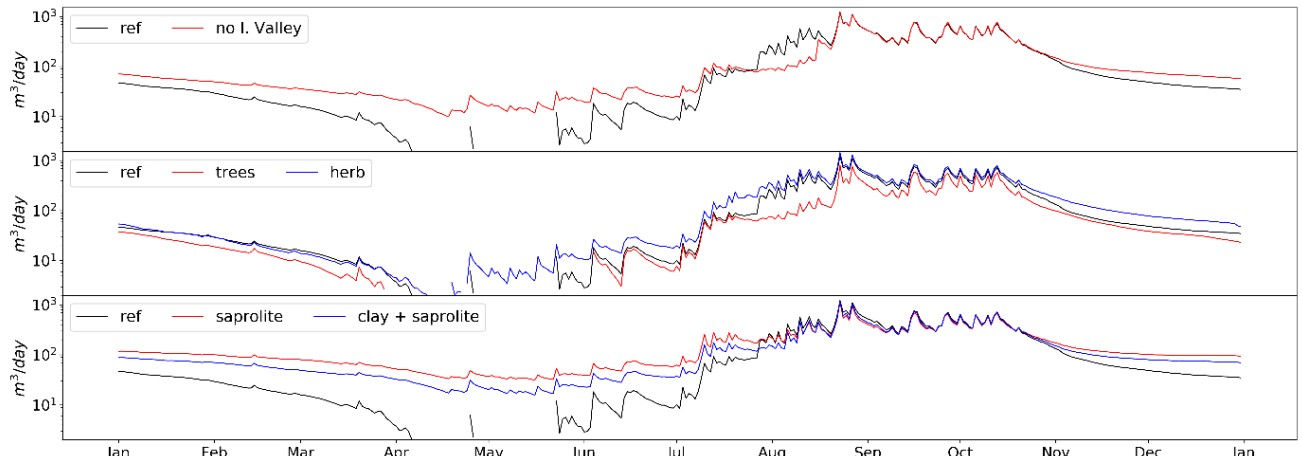

**Figure 9 : Simulated streamflow of year 2011 for the three virtual experiments. The reference run is always displayed as a black line. In the lower panel, the clay+saprolite refers to the 'mixed' case where H3 layer is made of clayey material on the left bank and saprolite material on the right bank.**

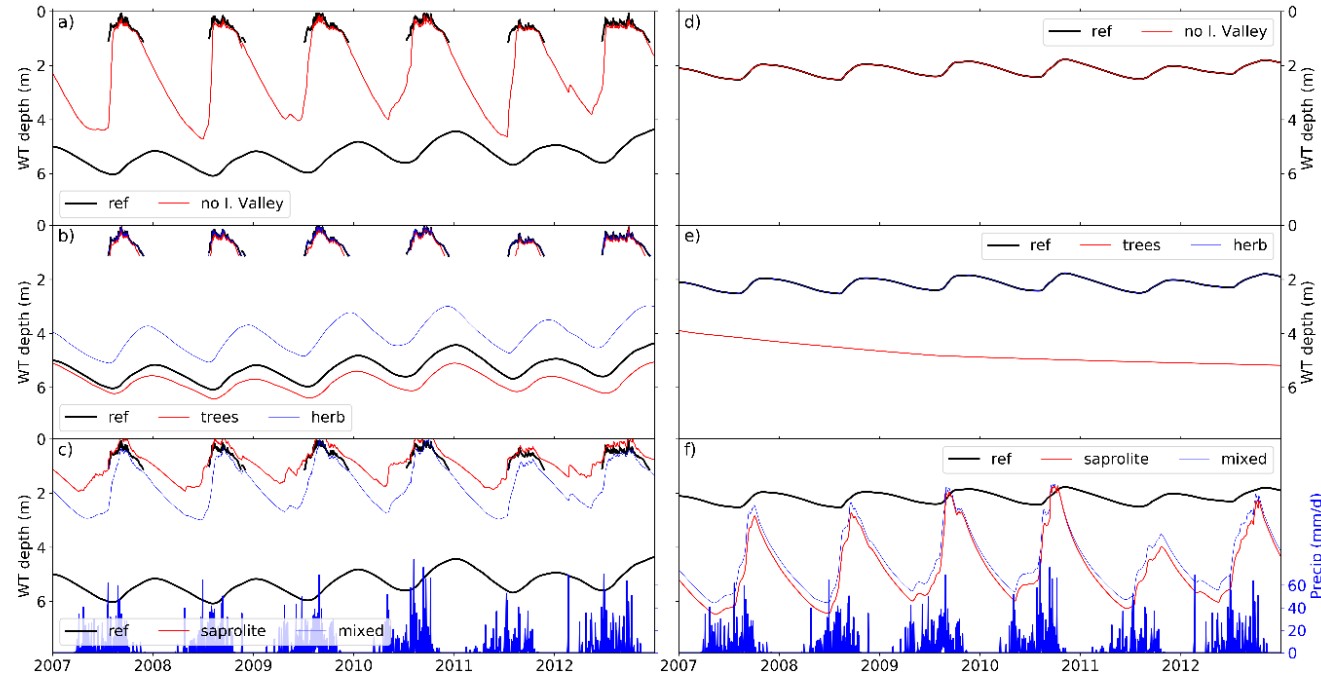

**Figure 10: Simulated water table in the valley thalveg (left) at P1 and on the upper contributive area (right) at P2**