# Peer review of "Hydrological functioning of West-African inland valleys explored with a critical zone model"

_Hydrology and Earth System Sciences, 2018_

## Referee Comment (RC1) · Anonymous Referee #1 · 19 Jun 2018

Hess-2018-219

Hector et al present a synthesis of observations and modeling to better understanding water resources in West-Africa. The work is highly relevant to the interests of the readership of HESS and this group and the AMMA-CATCH project have a long history studying this region. I found this work to be well written, well organized and novel. I have a special appreciation for how well the model and observations agree, how extensive the datasets are and how hard it is to work in this region. I have some comments below which i feel will improve the quality and clarity of this work but in general I find it suitable for publication in HESS with minor revisions.

minor comments:

[Figure]

1. The experiments with low-K layers were interesting and provided some counter-intuitive findings. Did the authors conduct any experiments where the layers were discontinuous? If not, can they comment about how this might impact results?

2. Domain geometry. Did the authors experiment with other domain configurations? It would be interesting to see how changes in slope for the v-catchment "banks" changed response. Do the choices in slope represent something in the real catchment?

3. Boundary conditions. Perhaps I missed it, but how were the boundaries set for the tilted-V catchment? Did this have an impact on e.g. water table dynamics and fluxes?

typographical comments:

p3 line 23, double parentheses

p5 line 32, double parentheses

p4. line 29, double parentheses– issue with a reference manager?

p2. line 15, URL better as a footnote than inline citation

p6, line 27, Manning's EQ

---

## Referee Comment (RC2) · Anonymous Referee #2 · 19 Jun 2018

**Review of "Hydrological functioning of West-African inland valleys explored with a critical zone model" by Hector et al. (2018)**

**General comments**

This manuscript studies the hydrology of an inland valley in West Africa. The impact of LULC changes on such hydrological systems has also been assessed. This work is of interest to the broad readership of HESS. The manuscript is well-written. The authors have also made a commendable effort of using observations to validate the model results. However, there are some issues that must be addressed before publication (please see the specific comments section below).

**Specific comments**

The manuscript aims to answer the following questions: 1) what are the main characteristics of the hydrological functioning of inland valleys in the Sudanian area of West Africa? 2) What is the impact of LULC changes of such systems? In order to answer these questions, the authors have presented simulations from a tilted-v catchment, which is forced using the atmospheric variables from the AMMA-CATCH observatory. The subsurface hydrogeology of the area has been represented using data from the previous literature. The tilted-v catchment is being used as the representative of the real catchment area in West Africa. However, it is not clear from the manuscript how the slopes of the tilted-v catchment (4% and 2% in Y- and X-directions, respectively) are representative of the real topography? This is important because the authors are using ParFlow, which is a 3D variably saturated surface water/groundwater flow model. The water flow in this model is dominated by topography.

The authors presented the comparisons between observations and simulation results. How much does the representation of the topographic slopes contribute to the differences between the observed and simulated fluxes? Moreover, without a comprehensive representation of the real topography or justification of the considered topographic slopes, how can this be assured that the hydrological processes of the simulated tilted-v are representative of those of the actual catchment? Reasonable agreements between the observations and simulation results do not answer all these questions. Is it worthwhile to consider the real topography (which is very possible as ParFlow-CLM is a distributed model) of the area rather than a tilted-v catchment with arbitrary slopes? These issues should be clarified in the manuscript before publication.

---

## Editor Comment (EC1) · E. Zehe (Editor) · 22 Jun 2018

Dear Dr. Hector,

within the spirit of HESS I'd like to encourage you to respond the postet reviews. Please note that HESS does at this stage not expect a full rebutal. The core idea of the discussion phase is to provide the authors and the reviewers for a debate.

Best regards,

Erwin Zehe

---

## Referee Comment (RC3) · Anonymous Referee #3 · 4 Jul 2018

Hector et al. present an interesting study of an elementary catchment (what is elementary?) replicated from the Sudanian Savanna (please use this term instead of Sudanian area) which is based on a plethora of field data and experience from a previous study. The focus are inland valleys and their hydrologic response and sensitivity to hydrogeologic heterogeneity and land cover change. The model is validated against the field data, which lends confidence in the results. This is a well-performed case study, which demonstrates that, in case of bulk mass fluxes, heterogeneity may play a minor role in the overall valley hydrologic response. Key are the boundary conditions and the interactions with the land surface.

I have only two points that the authors should address and some minor comments below. First, the authors state that this is a synthetic study of an elementary catchment,

however, which is based on their best knowledge and a plethora of data of the real system. In addition, the model is validated quite comprehensively. This is not consistent. In my opinion, the authors present a well-informed case study and not a synthetic study, which needs to be made clear in the text. Second, if they seek generalization of their results beyond their test site (and I feel that is what they suggest with the term elementary catchment) they need to find a way to make the results transferable and show this by applying their findings to inland valley across the region. In addition, the manuscript will benefit strongly from a careful check of language and grammar.

Minor comments Define the length of the time series earlier in the text. Provide information the climatology of the region at the beginning. The forcing is homogeneous over the model domain? Is it evaluation or validation data; please check section title. If you validate with data, why is this a synthetic modeling study? I would prefer Sudanian Savanna instead of Sudanian area. Remove "..." from enumerations. I prefer all variable symbols ($Q$, $ET$, etc.) in italic. In the EOF analyses, I assume that additional modes do not show any useful information, which I find disappointing, because I would have expected perhaps also some variance over longer time scales in storages in the 7 year time series. The annual signal is clear, which is not surprising.

---

## Author Comment (AC1) · 4 Jul 2018

Thank you for all the comments and questions regarding to our submission to HESS: Hydrological functioning of West-African inland valleys explored with a critical zone model".

Both two reviewers (#1 & #2) had relevant interrogations about the impact of topography on the generalization of the results presented in this study.

Although we specified that the V-shaped (virtual) catchment slopes correspond to the actual slopes of the reference inland valley of Nalohou (L. 14: "(mean N-S and E-W slopes correspond to those imposed on the virtual V-shaped catchment)"), it is clear that more details should be provided.

[Figure]

Nalohou catchment has been surveyed with double-difference GPS campaigns to provide accurate topography at 5m resolution. Local mean N-S & E-W slopes come from such dataset.

At the Upper Oueme catchment scale (14 000 km2), comprising many more inland valleys (Giertz et al., 2012), a quick slope analysis from 3" (about 90m) Hydrosheds DEM provided an average slope of 2.6 %. On nearby Togo in similar geomorpholigical context (Région centrale, 13000 km2), Runge (1991), noted that 80% of surface have slopes comprised in the range 1.7 – 5.2 %. They further identified inland valley to develop preferentially in the range 3.0 – 4.4 % along the steepest axis. The transverse axis (X in our case) ranges from 1.7 to 3.5% and the longitudinal axis (Y) from 3.5 to 5.2%. These numbers are very close to our virtual catchment (X: 2% Y: 4%).

Despite this relevance with respect to other (including larger scales) inland valleys, we wanted to check the impact of slopes on simulation results (budgets, and agreement to observations, as suggested by reviewers. We have not had the time to conduct the virtual experiments so far but could do so if needed.

As for the other comments of reviewer #1: - No-flow boundary conditions where applied over the catchment sides (as stated L. 22), but we did not precise that a no-flow boundary condition is also applied to the bottom of the catchment. We could add this precision. This choice is backed-up by the limited regional gradient as explained by 1) the low fracture connectivity within the bedrock (El Fahem et al., 2008) and 2) the low transverse hydraulic gradient at the larger scale (gradient of 0.01 m/m over about 3km) showing very little variations between the dry and the wet season, and suggesting limited larger-scale transverse flow which would affect the boundary conditions. These numbers are based on Hama Garba's Msc thesis (Hama Garba, 2016), available upon request.

- concerning the potential impact of discontinuous layers, this would indeed be an interesting study in itself. There is an extensive literature on the topic, including with

the same model (Atchley & Maxwell, 2011, Meyerhoff & Maxwell, 2011, Meyerhoff et al., 2014, Gilbert et al., 2016, among others), which we would could easily add to the paper with addititional comments.

Many thanks again with the reviewing work,

The authors.

References:

Atchley, A. L. and Maxwell, R. M.: Influences of subsurface heterogeneity and vegetation cover on soil moisture, surface temperature and evapotranspiration at hillslope scales, Hydrogeol J, 19(2), 289–305, doi:10.1007/s10040-010-0690-1, 2011.

El-Fahem, T.: Hydrogeological conceptualisation of a tropical river catchment in a crystalline basement area and transfer into a numerical groundwater flow model. Case study for the upper Ouémé catchment in Benin, Thèse de doctorat, Université de Bonn. [online] Available from: http://hss.ulb.uni-bonn.de/2008/1509/1509-1.pdf (Accessed 3 February 2014), 2008.

Giertz, S., Steup, G. and Schönbrodt, S.: Use and constraints on the use of inland valley ecosystems in central Benin: results from an inland valley survey, Erdkunde, 66(3), 239–253, 2012.

Gilbert, J. M., Jefferson, J. L., Constantine, P. G. and Maxwell, R. M.: Global spatial sensitivity of runoff to subsurface permeability using the active subspace method, Advances in Water Resources, 92, 30–42, doi:10.1016/j.advwatres.2016.03.020, 2016.

Hama Garba, O. K.: Etude Hydrogéologique et Géophysique du bassin versant de l'Ara au Benin, Memoire de Master, Université d'Abomey-Calavi, Université d'Abomey-Calavi., 2016.

Meyerhoff, S. B. and Maxwell, R. M.: Quantifying the effects of subsurface heterogeneity on hillslope runoff using a stochastic approach, Hydrogeol J, 19(8), 1515–1530,

doi:10.1007/s10040-011-0753-y, 2011.

Meyerhoff, S. B., Maxwell, R. M., Graham, W. D. and Williams, J. L.: Improved hydrograph prediction through subsurface characterization: conditional stochastic hillslope simulations, Hydrogeol J, 22(6), 1329–1343, doi:10.1007/s10040-014-1112-6, 2014.

Runge , J. (1991): Geomorphological depressions (Bas-fonds) and present-day erosion processes on the planation surface of Central-Togo/Westafrica. In: Erdkunde 45, 52-65. DOI: 10.3112/erdkunde.1991.01.05

---

## Referee Comment (RC4) · Anonymous Referee #4 · 15 Jul 2018

Review hess-2018-219

General Interesting paper, formalizing a complex set of hydrological processes. The work is framed within the Critical Zone approach.

Main comment With the information provided, it is basically impossible to reproduce any results. There are some tables with parameters but it would be so much easier if the complete code and data were made available, preferably as a Docker container. Where are the the topographic data? Where are the measurement values? Which version of ParFlow was used (why not link to the code?)? See, for example, the article "Most computational hydrology is not reproducible, so is it really science?" (https://doi.org/10.1002/2016WR019285). This is a serious issue within hydrology and I would urge the authors to spend some time carefully curating their code and data.

[Figure]

This would greatly improve usability and uptake of their approach.

Minor comment It is stated that a complete study would be needed to describe the effects of H2. I agree but perhaps tell the reader a bit more about the geology. Is the geology metamorphic or granitic? The work by Bertrand may be useful here with a nice overview presented in Sitapha Diatta's work on impermeable layers in inland valleys (http://docnum.univ-lorraine.fr/public/SCD_T_1996_0043_DIATTA.pdf). Being familiar with their work, I can still not determine if the Nalohou is similar or completely different. So please add some morphological/geological information.

————————————————————

---

## Author Comment (AC2) · 17 Jul 2018

We would like to thank again the four reviewers who produced significant comments and raised important issues that we hope to address convincingly in order to strengthen the paper.

- On the slopes issues: going further than our previous reply, we ran two more simulations using the reference case, and multiplying the slopes by a factor 2 and a factor 0.5. The results are given below in terms of mean + standard deviations over the 6 years period for the different budget components. Other test cases are presented for comparison.

[ table is shown in the attached material]

[Figure]

Table 1: summary of the annual average and standard deviation for the 6 simulated years and for each budget component. Reference case with slopes multiplied by 0.5 and 2 are also shown.

There is an insignificant impact on the interannual variability (standard deviation), but a significant impact on yearly averages. The slope values taken in this short experiment are extreme cases as compared to the regional topography where inland-valleys are found (see previous comment). Yet their impact is comparable to the impact of vegetation distribution (trees case $\sim$ slope ref x0.5 ; herb case $\sim$ slope ref x2). Lower slopes (resp. higher) decrease (resp. increase) lateral transfers to the benefit (resp. cost) of evapotranspiration. This short analysis can be added to the paper, together with the references given in our previous reply to comments.

The goal of the paper is not to reproduce absolutely the behavior of our benchmark catchment say, by using the real topography, as we want to derive the controlling factors of inland valley critical zone systems -through virtual experiments- among factors that are either susceptible to evolve (land cover), that characterize inland valleys (thalweg clay lens) or are still largely unknown (subsurface lithology). To this respect, slopes are not unknowns nor susceptible to evolve much (as land cover which have been observed to change significantly, and are also projected to do so) and are not targets of this study. However, catchment slopes may significantly impact the water budgets as shown by these two experiments, and impact our ability to generalize the inland valley functioning based on this study only.

- On the synthetic vs case study issue (reviewer #3): We indeed used a "well-informed case study", from which we drew schematic configurations to build virtual experiments, also based on our knowledge and litterature review of other inland valleys configurations. From these results we draw conclusions on the sensitivity of such critical zone system to its unknown and likely-to-evolve components (vegetation, clay lens, lithology). We will add some lines in the introduction, abstract & conclusion to clarify this approach.

- On the study transferability to other inland valleys (reviewer #3):

The modeled synthetic case is built based on our knowledge of inland valleys. The model is not intended to reproduce every single inland valley behavior, but instead draw the first order lines of the impact of main inland valleys characteristics (clay lens), of their sensitivity to changes (in land cover) and to largely unknowns parameters (sus-bsurface lithology). We reckon that further explanations should be added to the introduction, and we will do so.

- On the reproducibility of the study (reviewer #4):

Thank you for pointing that out. We will indeed attach the main ParFlow configuration script, and link to the ParFlow code. We will investigate the feasibility of a Docker container for the PF environment.

Minor comments of reviewer #3:

- 'elementary' refers to the usual term : the smallest catchment (two hillslopes) draining into first order streams. We will add this precision.

- The meteorological forcing is indeed homogeneous over the domain. The domain extent is small enough to justify this. We will mention this precision in the text, thank you!

- The data is evaluation data (as in section 2.5 title): we will replace the single occurrence of 'validation' by evaluation because this is indeed evaluation of a synthetic case. Thanks much for pointing that out.

- We will replace Sudanian area by Sudanian Savanna.

- On the EOF modes: Mode 2 accounts for about 8% of the total variance, and this is why we did not discuss it. However, you are right, the interanual storage variations are present in this mode, but together with some residuals at the seasonal level. As our hybrid gravity data only covers 2 years, which is limited for interanual terms compari-

son, and because of the low level of variance explained (there is much more signal at the seasonal level than the interanual one), we did not choose to show other modes than mode 1. We will add a line to mention the content of mode 2, as explained above.

Minor comments of reviewer #4:

-the Geology is metamorphic, thank you for pointing out this omission.

- Many thanks also for the reference given, which we did not know, but significantly strengthen the current study. This inland valley study describes the same hydrological behavior between the perched water table and the permanent water table : a disconnexion in the valley thalweg and some lateral connexion upstream, as described in Figure 13 in Hector, B., Séguis, L., Hinderer, J., Cohard, J.-M., Wubda, M., Descloitres, M., Benarrosh, N. and Boy, J.-P.: Water storage changes as a marker for base flow generation processes in a tropical humid basement catchment (Benin): Insights from hybrid gravimetry, Water Resour. Res., doi:10.1002/2014WR015773, 2015.

We will discuss our settings and results with respect to their study.

Please also note the supplement to this comment:
https://www.hydrol-earth-syst-sci-discuss.net/hess-2018-219/hess-2018-219-AC2-supplement.pdf

---

## Author Comment (AC3) · 3 Aug 2018

Thank you for all the comments and questions regarding to our submission to HESS: Hydrological functioning of West-African inland valleys explored with a critical zone model". Many thanks to the four reviewers who raised relevant questions that we hope to address in this final response. We will mainly gather previous replies written in the debate phase and add our suggestions for changes in the manuscript.

RV#1 1. The experiments with low-K layers were interesting and provided some counter-intuitive findings. Did the authors conduct any experiments where the layers were discontinuous? If not, can they comment about how this might impact results?

=> Concerning the potential impact of discontinuous layers, this would indeed be an

interesting study in itself. There is an extensive literature on the topic, including with the same model (Atchley & Maxwell, 2011, Meyerhoff & Maxwell, 2011, Meyerhoff et al., 2014, Gilbert et al., 2016, among others). The target of the study was essentially to not focus on local heterogeneity to extract the main features and main behaviors of inland valleys.

However, there are still some discussion points that we should include in the paper: - The have noted the presence of dual-porosity soils, which we do not take into account in this study. It is likely that the dampening effect of the unsaturated clays contribute to play this role. - The valley thalweg is oversimplified, and the lack of recharge of the permanent water table by direct infiltration (see Fig. 4.a), is probably due to the homogeneity of the valley thalweg material, while in the real world, heterogeneity and clay lens topography certainly impact the spatial distribution of recharge, as explained p.12 l.19-22. - Figure 4.a and 4.b belong to different simulations, although both associated data are real world data of the same catchment. This means that we actually still miss some features: if we set a 'higher' permeability to the valley banks, it will result in the filling of the permanent water table in the valley thalweg, which is not observed. To successfully match the observations, we would definitely have to include lateral variations in permeability. But again, this was not the target of this study.

Modifications in the manuscript:

P. 5 L 7.: 'There is an extensive literature, using the same critical zone modeling framework that provides insights on the effect of lateral variability of soil properties through dedicated sensitivity studies (Atchley & Maxwell, 2011, Meyerhoff & Maxwell, 2011, Meyerhoff et al., 2014, Gilbert et al., 2016).'

P19 L 21: Furthermore, one should note that Fig. 4.a and 4.b show results from different simulations (reference and saprolite, respectively), although both associated water table levels are real world data of the same catchment. This means that some features are still missing in order to comprehensively reproduce the behaviors of this

specific catchment. If a high permeability is set to the valley banks (as in the saprolite simulation), it will result in the filling of most of the valley thalweg by the permanent water table and the absence of the perched water table, which is not observed. On the opposite, in the low permeability scenario (reference simulation), the simulated water table on the valley banks mismatches the observed data. To successfully match all the observations, lateral variations in the permeability are needed, but this was not the target of this study, which was instead the extraction of main features and behaviors of inland valleys.

Added references Atchley, A. L. and Maxwell, R. M.: Influences of subsurface heterogeneity and vegetation cover on soil moisture, surface temperature and evapotranspiration at hillslope scales, Hydrogeol J, 19(2), 289–305, doi:10.1007/s10040-010-0690-1, 2011.

Gilbert, J. M., Jefferson, J. L., Constantine, P. G. and Maxwell, R. M.: Global spatial sensitivity of runoff to subsurface permeability using the active subspace method, Advances in Water Resources, 92, 30–42, doi:10.1016/j.advwatres.2016.03.020, 2016.

Meyerhoff, S. B. and Maxwell, R. M.: Quantifying the effects of subsurface heterogeneity on hillslope runoff using a stochastic approach, Hydrogeol J, 19(8), 1515–1530, doi:10.1007/s10040-011-0753-y, 2011.

Meyerhoff, S. B., Maxwell, R. M., Graham, W. D. and Williams, J. L.: Improved hydrograph prediction through subsurface characterization: conditional stochastic hillslope simulations, Hydrogeol J, 22(6), 1329–1343, doi:10.1007/s10040-014-1112-6, 2014.

2. Domain geometry. Did the authors experiment with other domain configurations? It would be interesting to see how changes in slope for the v-catchment "banks" changed response. Do the choices in slope represent something in the real catchment?

=> As stated P.9 L14-15, 'topographic slopes are gentle (mean N-S and E-W slopes correspond to those imposed on the virtual V-shaped catchment).'. This comment is

shared with RV2, and we will provide more response below (in final response to RV2), together with manuscript changes suggestions.

3. Boundary conditions. Perhaps I missed it, but how were the boundaries set for the tilted-V catchment? Did this have an impact on e.g. water table dynamics and fluxes?

=> No-flow boundary conditions where applied over the catchment sides (as stated P. 7 L. 22), but we did not precise that a no-flow boundary condition is also applied to the bottom of the catchment. We could add this precision. This choice is backed-up by the limited regional gradient as explained by 1) the low fracture connectivity within the bedrock (El Fahem et al., 2008) and 2) the low transverse hydraulic gradient at the larger scale (gradient of 0.01 m/m over about 3km) showing very little variations between the dry and the wet season, and suggesting limited larger-scale transverse flow which would affect the boundary conditions. These numbers are based on Hama Garba's Msc thesis (Hama Garba, 2016), available upon request.

Modifications in the manuscript: - P7. L-22: 'No-flow boundary conditions are applied on the catchment edges and bottom.

---

## Author Comment (AC4) · 3 Aug 2018

Thank you for all the comments and questions regarding to our submission to HESS: Hydrological functioning of West-African inland valleys explored with a critical zone model". Many thanks to the four reviewers who raised relevant questions that we hope to address in this final response. We will mainly gather previous replies written in the debate phase and add our suggestions for changes in the manuscript.

**RV#2**
**It is not clear from the manuscript how the slopes of the tilted-v catchment (4% and 2% in Y- and X-directions, respectively) are representative of the real topography? This is important because the authors are using ParFlow, which is a 3D variably saturated surface water/groundwater flow model. The water flow in this model is dominated by topography. The authors presented the comparisons between observations and simulation results. How much does the representation of the topographic slopes contribute to the differences between the observed and simulated fluxes? Moreover, without a comprehensive representation of the real topography or justification of the considered topographic slopes, how can this be assured that the hydrological processes of the simulated tilted-v are representative of those of the actual catchment? Reasonable agreements between the observations and simulation results do not answer all these questions. Is it worthwhile to consider the real topography (which is very possible as ParFlow-CLM is a distributed model) of the area rather than a tilted-v catchment with arbitrary slopes? These issues should be clarified in the manuscript before publication.**

=> Both two reviewers (#1 & #2) had relevant interrogations about the impact of topography on the generalization of the results presented in this study.

Although we specified that the V-shaped (virtual) catchment slopes correspond to the actual slopes of the reference inland valley of Nalohou (P.9 L. 14: "(mean N-S and E-W slopes correspond to those imposed on the virtual V-shaped catchment)"), it is clear that more details should be provided.

Nalohou catchment has been surveyed with double-difference GPS campaigns to provide accurate topography at 5m resolution. Local mean N-S & E-W slopes come from such dataset.

At the Upper Oueme catchment scale (14 000 km2), comprising many more inland valleys (Giertz et al., 2012), a quick slope analysis from 3" (about 90m) Hydrosheds DEM provided an average slope of 2.6 %. On nearby Togo in similar geomorpholigical context (Région centrale, 13000 km2), Runge (1991), noted that 80% of surface have slopes comprised in the range 1.7 – 5.2 %. They further identified inland valley to develop preferentially in the range 3.0 – 4.4 % along the steepest axis. The transverse axis (X in our case) ranges from 1.7 to 3.5% and the longitudinal axis (Y) from 3.5 to 5.2%. These numbers are very close to our virtual catchment (X: 2% Y: 4%).

Despite this relevance with respect to other (including larger scales) inland valleys, we wanted to check the impact of slopes on simulation results (budgets, and agreement to observations, as suggested by reviewers. We have not had the time to conduct the virtual experiments so far but could do so if needed.

going further than our previous reply, we ran two more simulations using the reference case, and multiplying the slopes by a factor 2 and a factor 0.5. The results are given below in terms of mean + standard deviations over the 6 years period for the different budget components. Other test cases are presented for comparison.

|  | ET (mm) | Q (mm) | S (mm) |
|---|---|---|---|
| reference | 839 ± 64 | 454 ± 151 | 8 ± 26 |
| No Inland valley | 830 ± 64 | 463 ± 153 | 8 ± 24 |
| trees | 944 ± 53 | 361 ± 144 | -5 ± 54 |
| herb | 791 ± 72 | 500 ± 151 | 10 ± 17 |
| saprolite | 793 ± 60 | 495 ± 153 | 13 ± 36 |
| mixed | 816 ± 62 | 475 ± 154 | 10 ± 28 |
| **Slope ref x 0.5** | **933 ± 62** | **354 ± 154** | **14 ± 28** |
| **Slope ref x 2** | **773 ± 64** | **526 ± 157** | **1 ± 21** |

Table 1: summary of the annual average and standard deviation for the 6 simulated years and for each budget component. Reference case with slopes multiplied by 0.5 and 2 are also shown.

There is an insignificant impact on the interannual variability (standard deviation), but a significant impact on yearly averages. The slope values taken in this short experiment are extreme cases as compared to the regional topography where inland-valleys are found (see previous comment). Yet their impact is comparable to the impact of vegetation distribution (trees case ~ slope ref x0.5 ; herb case ~ slope ref x2). Lower slopes (resp. higher) decrease (resp. increase) lateral transfers to the benefit (resp. cost) of evapotranspiration. This short analysis can be added to the paper, together with the references given in our previous reply to comments.

The goal of the paper is not to reproduce absolutely the behavior of our benchmark catchment say, by using the real topography, as we want to derive the controlling factors of inland valley critical zone systems -through virtual experiments- among factors that are either susceptible to evolve (land cover), that characterize inland valleys (thalweg clay lens) or are still largely unknown (subsurface lithology). To this respect, slopes are not unknowns nor susceptible to evolve much (as land cover which have been observed to change significantly, and are also projected to do so) and are not targets of this study. However, catchment slopes may significantly impact the water budgets as shown by these two experiments, and impact our ability to generalize the inland valley functioning based on this study only.

**Modifications in the manuscript:**

P5. L.6. add a section: 2.1.2: Topography:
At the Upper Oueme catchment scale (14 000 km2, which includes the Nalohou area), comprising many inland valleys (Giertz et al., 2012), a slope analysis from 3" (about 90m) Hydrosheds DEM provided an average slope of 2.6 %. On nearby Togo in similar geomorphological context (Région centrale, 13000 km2), Runge (1991), noted that 80% of surface have slopes comprised in the range 1.7 – 5.2 %. They further identified inland valleys to develop preferentially in the range 3.0 – 4.4 % along the steepest axis. The transverse axis ranges from 1.7 to 3.5% and the longitudinal axis from 3.5 to 5.2%.  For the specific Nalohou catchment, mean longitudinal slope (N-S) is 4% and mean transverse (E-W) absolute slope is 2%, according to a 5m lateral resolution DEM derived from DGPS survey, as described in Hector et al., (2013).

P11. L24. New paragraph:
Furthermore, as topography drives lateral flow, we conducted two more experiments on the impact of slopes, using the reference case, and multiplying the slopes by a factor 2 and a factor 0.5. These simplified experiments allow to extract the effect of extreme topography range found in this environment (Runge, 1991).

P 15. L 6. New paragraph:
For the simplified slope experiments, there is an insignificant impact on the interannual variability (standard deviation), but a significant impact on yearly averages. The slope values taken in this short experiment are extreme cases as compared to the regional topography where inland-valleys are found. Yet their impact is comparable to the impact of vegetation distribution (trees case ~ slope ref x0.5 ; herb case ~ slope ref x2). Lower slopes (resp. higher) decrease (resp. increase) lateral transfers to the benefit (resp. cost) of evapotranspiration.

Modification in Table 6:

|  | ET (mm) | Q (mm) | S (mm) |
| --- | --- | --- | --- |
| reference | 839 ± 64 | 454 ± 151 | 8 ± 26 |
| No Inland valley | 830 ± 64 | 463 ± 153 | 8 ± 24 |
| trees | 944 ± 53 | 361 ± 144 | -5 ± 54 |
| herb | 791 ± 72 | 500 ± 151 | 10 ± 17 |
| saprolite | 793 ± 60 | 495 ± 153 | 13 ± 36 |
| mixed | 816 ± 62 | 475 ± 154 | 10 ± 28 |
| **Slope ref x 0.5** | **933 ± 62** | **354 ± 154** | **14 ± 28** |
| **Slope ref x 2** | **773 ± 64** | **526 ± 157** | **1 ± 21** |

P.17. L7. New paragraph:

The goal of the paper is to derive the controlling factors of inland valley critical zone systems -through virtual experiments- among factors that are either susceptible to evolve (land cover), that characterize inland valleys (thalweg clay lens) or are still largely unknown (subsurface lithology). To this respect, slopes and topographic effects are not unknowns nor susceptible to evolve much (as land cover which have been observed to change significantly, and are also projected to do so) and are not targets of this study. However, catchment slopes may significantly impact the water budgets as shown by the two experiments (Table 6), and impact our ability to generalize the inland valley functioning based on this study only.

Added references

Giertz, S., Steup, G. and Schönbrodt, S.: Use and constraints on the use of inland valley ecosystems in central Benin: results from an inland valley survey, Erdkunde, 66(3), 239–253, 2012.

Runge , J. (1991): Geomorphological depressions (Bas-fonds) and present-day erosion processes on the planation surface of Central-Togo/Westafrica. In: Erdkunde 45, 52-65. DOI: 10.3112/erdkunde.1991.01.05

---

## Author Comment (AC5) · 3 Aug 2018

Thank you for all the comments and questions regarding to our submission to HESS: Hydrological functioning of West-African inland valleys explored with a critical zone model". Many thanks to the four reviewers who raised relevant questions that we hope to address in this final response. We will mainly gather previous replies written in the debate phase and add our suggestions for changes in the manuscript.

RV #3 1. The authors state that this is a synthetic study of an elementary catchment, however, which is based on their best knowledge and a plethora of data of the real system. In addition, the model is validated quite comprehensively. This is not consistent. In my opinion, the authors present a well-informed case study and not a synthetic

study, which needs to be made clear in the text. If [the authors] seek generalization of their results beyond their test site (and I feel that is what they suggest with the term elementary catchment) they need to find a way to make the results transferable and show this by applying their findings to inland valley across the region.

=> In the text, the two occurrences of 'synthetic' refer to a generic virtual elementary catchment comprising an inland valley (as in P3. L18: 'idealized elementary V-shaped catchment'), designed for generalization, and regardless of the "well-informed case study", which we use to draw schematic configurations to build virtual experiments, also based on our knowledge and literature review of other inland valleys configurations. From these results we draw conclusions on the sensitivity of such critical zone system to its unknown and likely-to-evolve components (vegetation, clay lens, lithology). We will remove occurrences of 'synthetic', which are misleading.

'Elementary' refers to the usual term : the smallest catchment (two hillslopes) draining into first order streams. The modeled synthetic case is built based on our knowledge of inland valleys. The model is not intended to reproduce every single inland valley behavior, but instead draw the first order lines of the impact of main inland valleys characteristics (clay lens), of their sensitivity to changes (in land cover) and to largely unknowns parameters (susbsurface lithology). We reckon that further explanations should be added to the abstract, introduction, and conclusion. We also hope that the references added on topography (see reply to reviewer #1 & #2 above), and on similar hydrological behavior from the inputs of reviewer #4 (see below) further help the understanding of the framework.

Modifications in the manuscript: Abstract. L17: replace 'synthetic' by 'virtual generic' Abstract. L.17: Model forcings are based on 20 years data from the AMMA-CATCH observation service and parameters are evaluated against multiple field data (Q, evapotranspiration –ET-, soil moisture, water table levels, and water storage) acquired on a specific elementary catchment. The hydrological model applied to the conceptual lithological/pedological model proposed in this study reproduces the main behaviors

observed on a highly instrumented elementary catchment., allowing to further conduct virtual experiments:

P3. L19: replace: 'This deterministic modeling approach largely builds on the large panel of observations available within the AMMA-CATCH observation service (www.amma-catch.org (Galle et al., Submitted; Lebel et al., 2009) and several campaigns, as well as on a highly instrumented elementary catchment for which we previously built a conceptual lithological/pedological model (Hector et al., 2015).' In the first section, we briefly discuss the physical environment and how we model it using a physically-based CZ model. Then, we present the results of a reference case, which are compared to observations from an elementary headwater catchment in the hard-rock area of the Sudanian region, to show that the model is able to reproduce to a large extent the complex critical zone behavior. We finally use the results of a set a virtual experiments to infer the model sensitivity to the main inland valley features (presence of a clay layer, hydrodynamic properties of the contributive areas, vegetation distribution), and discuss these results.

by

' This idealized elementary V-shaped catchment is built based on the main features of elementary catchments comprising an inland valley in the region (slope, clay lens, vegetation distribution), but also following a highly instrumented elementary catchment for which we previously built a conceptual lithological/pedological model (Hector et al., 2015). In the first section, we briefly discuss the physical environment and how we model it over a 7-years period using a physically-based CZ model. Then, we present the results of a single simulation, which are compared to observations from an elementary headwater catchment in the hard-rock area of the Sudanian region, available within the AMMA-CATCH observation service (www.amma-catch.org (Galle et al., Submitted; Lebel et al., 2009) and several campaigns, to show that the model is able to reproduce to a large extent the complex critical zone behavior. We finally use the results of a set of virtual experiments to infer the model sensitivity to the main inland valley features

(presence of a clay layer in the valley thalweg and hydrodynamic properties of the contributive areas) and the vegetation distribution, likely to evolve, and discuss these results.

P.4 L4: ...built following the literature of the region and on inland valleys, and the...

P7. L13:Y-direction slope is 4 % and X-direction slope is 2 %, following Runge (1991) and specific values for the Nalohou catchment.

P19. L23: replace: 'In this paper, we studied the hydrological functioning of Sudanian inland valleys and their sensitivity to land cover and contributive areas through deterministic sensitivity experiments using a physically-based critical zone (CZ) model applied on a synthetic catchment which comprises an inland valley. This is a first approach to try to investigate what can control and explain the behavior of an inland valley.'

By: 'In this paper, we studied the hydrological functioning of Sudanian inland valleys and their sensitivity to land cover and their main features (pedology of contributive areas, clay lens), through deterministic sensitivity experiments using a physically-based critical zone (CZ) model applied on a virtual generic catchment which comprises an inland valley. This is a first approach to try to investigate what can control and explain the behavior of an inland valley.'

3.Define the length of the time series earlier in the text. => Done in the introduction: see modification P3. L19

4. The forcing is homogeneous over the model domain? => The meteorological forcing is indeed homogeneous over the domain. The domain extent is small enough to justify this. We will mention this precision in the text, thank you! Modifications in the manuscript: P8. L15: PFCLM is being forced at a 30 mn time step, over the 7-years period 2006-2012, with spatially homogeneous forcings.

5. Is it evaluation or validation data; please check section title => The data is evaluation

data (as in section 2.5 title): we will replace the single occurrence of 'validation' by evaluation because this is indeed evaluation of a synthetic case. Thanks much for pointing that out.

7. I would prefer Sudanian Savanna instead of Sudanian area. => We will replace Sudanian area by Sudanian Savanna.

8. In the EOF analyses, I assume that additional modes do not show any useful information, which I find disappointing, because I would have expected perhaps also some variance over longer time scales in storages in the 7 year time series. The annual signal is clear, which is not surprising. => Mode 2 accounts for about 8% of the total variance, and this is why we did not discuss it. However, you are right, the interanual storage variations are present in this mode, but together with some residuals at the seasonal level. As our hybrid gravity data only covers 2 years, which is limited for interanual terms comparison, and because of the low level of variance explained (there is much more signal at the seasonal level than the interanual one), we did not choose to show other modes than mode 1. We will add a line to mention the content of mode 2, as explained above. Modifications in the manuscript: P12. L4. Mode 2 accounts for about 8% of the total variance, and although we do not discuss it in the current study, one should note that interannual storage variations together with some residuals at the seasonal level are present in this mode.

---

## Author Comment (AC6) · 3 Aug 2018

Thank you for all the comments and questions regarding to our submission to HESS: Hydrological functioning of West-African inland valleys explored with a critical zone model". Many thanks to the four reviewers who raised relevant questions that we hope to address in this final response. We will mainly gather previous replies written in the debate phase and add our suggestions for changes in the manuscript.

RV #4 1. With the information provided, it is basically impossible to reproduce any results. There are some tables with parameters but it would be so much easier if the complete code and data were made available, preferably as a Docker container. Where are the the topographic data? Where are the measurement values? Which

version of ParFlow was used (why not link to the code?)? See, for example, the article "Most computational hydrology is not reproducible, so is it really science?" (https://doi.org/10.1002/2016WR019285). This is a serious issue within hydrology and I would urge the authors to spend some time carefully curating their code and data. This would greatly improve usability and uptake of their approach.

=>Thank you for pointing that out. We will indeed attach the main ParFlow configuration script, as supplementary material (if this is relevant for the Editor) and link to the ParFlow code (there is a tagged version on PF Github that we will point to: https://github.com/basileh/parflow/releases/tag/v3.3.1-IGE). Input data will be made available upon request, together with a Docker image (too large to provide with the paper anyway) that allows to run the model in the same configuration as for the paper.

2. It is stated that a complete study would be needed to describe the effects of H2. I agree but perhaps tell the reader a bit more about the geology. Is the geology metamorphic or granitic? The work by Bertrand may be useful here with a nice overview presented in Sitapha Diatta's work on impermeable layers in inland valleys (http://docnum.univ-lorraine.fr/public/SCD_T_1996_0043_DIATTA.pdf). Being familiar with their work, I can still not determine if the Nalohou is similar or completely different. So please add some morphological/geological information.

=>The Geology is metamorphic, thank you for pointing out this omission. Many thanks also for the reference given, which we did not know, but significantly strengthen the current study. This inland valley study describes the same hydrological behavior between the perched water table and the permanent water table : a disconnexion in the valley thalweg and some lateral connexion upstream, as described in Figure 13 in Hector, B., Séguis, L., Hinderer, J., Cohard, J.-M., Wubda, M., Descloitres, M., Benarrosh, N. and Boy, J.-P.: Water storage changes as a marker for base flow generation processes in a tropical humid basement catchment (Benin): Insights from hybrid gravimetry, Water Resour. Res., doi:10.1002/2014WR015773, 2015. We will discuss our settings and results with respect to their study.

Modifications in the manuscript:

P4. L3: In this section, we briefly describe our conceptual representation of the main soil and vegetation characteristics found in the hard-rock basement areas of the Sudanian climatic region (yearly precipitation amount between 700 and 1400 mm), dominated by metamorphic settings.

P4. L11: this specific behavior is representative of inland valley functioning in similar geological/climatic context (see e.g. Brabant, 1991 in Cameroun or Diatta, 1996 in Ivory Coast)

References added: Diatta, S.: Les sols gris de bas versant sur granito-gneiss en région centrale de la Côte d'Ivoire : organisation toposéquentielle et spatiale, fonctionnement hydrologique : conséquences pour la riziculture, Nancy 1. [online] Available from: http://www.theses.fr/1996NAN10043 (Accessed 26 July 2018), 1996.